# GeoPhy: Differentiable Phylogenetic Inference via Geometric Gradients of Tree Topologies

**Takahiro Mimori**
Waseda University
RIKEN AIP
takahiro.mimori@aoni.waseda.jp

**Michiaki Hamada**
Waseda University
AIST-Waseda CBBD-OIL
Nippon Medical School
mhamada@waseda.jp

## Abstract

Phylogenetic inference, grounded in molecular evolution models, is essential for understanding the evolutionary relationships in biological data. Accounting for the uncertainty of phylogenetic tree variables, which include tree topologies and evolutionary distances on branches, is crucial for accurately inferring species relationships from molecular data and tasks requiring variable marginalization. Variational Bayesian methods are key to developing scalable, practical models; however, it remains challenging to conduct phylogenetic inference without restricting the combinatorially vast number of possible tree topologies. In this work, we introduce a novel, fully differentiable formulation of phylogenetic inference that leverages a unique representation of topological distributions in continuous geometric spaces. Through practical considerations on design spaces and control variates for gradient estimations, our approach, GeoPhy, enables variational inference without limiting the topological candidates. In experiments using real benchmark datasets, GeoPhy significantly outperformed other approximate Bayesian methods that considered whole topologies.

## 1 Introduction

Phylogenetic inference, the reconstruction of tree-structured evolutionary relationships between biological units, such as genes, cells, individuals, and species ranging from viruses to macro-organisms, is a fundamental problem in biology. As the phylogenetic relationships are often indirectly inferred from molecular observations, including DNA, RNA and protein sequences, Bayesian inference has been an essential tool to quantify the uncertainty of phylogeny. However, due to the complex nature of the phylogenetic tree object, which involve both a discrete topology and dependent continuous variables for branch lengths, the default approach for the phylogenetic inference has typically been an Markov-chain Monte Carlo (MCMC) method [29], enhanced with domain-specific techniques such as a mixed strategy for efficient exploration of tree topologies.

As an alternative approach to the conventional MCMCs, a variational Bayesian approach for phylogenetic inference was proposed in [42] (VBPI), which has subsequently been improved in the expressive powers of topology-dependent branch length distributions [39, 40]. Although these methods have presented accurate joint posterior distributions of topology and branch lengths for real datasets, they required reasonable preselection of candidate tree topologies to avoid a combinatorial explosion in the number of weight parameters beforehand. There have also been proposed variational approaches [24, 16] on top of the combinatorial sequential Monte Carlo method (CSMC[34]), which iteratively updated weighted topologies without the need for the preselection steps. However, the fidelity of the joint posterior distributions was still largely behind that of MCMC and VBPI as reported in [16].

37th Conference on Neural Information Processing Systems (NeurIPS 2023).

In this work, we propose a simple yet effective scheme for parameterizing a binary tree topological distribution with a transformation of continuous distributions. We further formulate a novel differentiable variational Bayesian approach named GeoPhy [1] to approximate the posterior distribution of the tree topology and branch lengths without preselection of candidate topologies. By constructing practical topological models based on Euclidean and hyperbolic spaces, and employing variance reduction techniques for the stochastic gradient estimates of the variational lower bound, we have developed a robust algorithm that enables stable, gradient-based phylogenetic inference. In our experiments using real biological datasets, we demonstrate that GeoPhy significantly outperforms other approaches, all without topological restrictions, in terms of the fidelity of the marginal log-likelihood (MLL) estimates to gold-standard provided with long-run MCMCs.

Our contributions are summarized as follows:

- We propose a representation of tree topological distributions based on continuous distributions, thereby we construct a fully differentiable method named GeoPhy for variational Bayesian phylogenetic inference without restricting the support of the topologies.

- By considering variance reduction in stochastic gradient estimates and simple construction of topological distributions on Euclidean and hyperbolic spaces, GeoPhy offers unprecedently close marginal log-likelihood estimates to gold standard MCMC runs, outperforming comparable approaches without topological preselections.

## 2  Background

### 2.1  Phylogenetic models

Let $\tau$ be represent an unrooted binary tree topology with $N$ leaf nodes (tips), and let $B_\tau$ denote a set of evolutionary distances defined on each of the branches of $\tau$. A phylogenetic tree $(\tau, B_\tau)$ represents an evolutionary relationship between $N$ species, which is inferred from molecular data, such as DNA, RNA or protein sequences obtained for the species. Let $Y = \{Y_{ij} \in \Omega\}_{1 \leq i \leq N, 1 \leq j \leq M}$ be a set of aligned sequences with length $M$ from the species, where $Y_{ij}$ denote a character (base) of the $i$-th sequence at $j$-th site, and is contained in a set of possible bases $\Omega$. For DNA sequences, $\Omega$ represents a set of 4-bit vectors, where each bit represents 'A', 'T', 'G', or 'C'. A likelihood model of the sequences $P(Y|\tau, B_\tau)$ is determined based on evolutionary assumptions. In this study, we follow a common practice for method evaluations [42, 40] as follows: $Y$ is assumed to be generated from a Markov process along the branches of $\tau$ in a site-independent manner; The base mutations are assumed to follow the Jukes-Cantor model [12]. The log-likelihood $\ln P(Y|\tau, B_\tau)$ can be calculated using Felsenstein's pruning algorithm [5], which is also known as the sum-product algorithm, and differentiable with respect to $B_\tau$.

### 2.2  Variational Bayesian phylogenetic inference

The variational inference problem for phylogenetic trees, which seeks to approximate the posterior probability $P(\tau, B_\tau|Y)$, is formulated as follows:

$$\min_Q D_{\mathrm{KL}} \left( Q(\tau)Q(B_\tau|\tau) \| P(\tau, B_\tau|Y) \right), \tag{1}$$

where $D_{\mathrm{KL}}$, $Q(\tau)$ and $Q(B_\tau|\tau)$ denote the Kullback-Leibler divergence, a variational tree topology distribution, and a variational branch length distribution, respectively. The first variational Bayesian phylogenetic inference method (VBPI) was proposed by Zhang and Matsen IV [42], which has been successively improved for the expressiveness of $Q(B_\tau|\tau)$ [39, 40]. For the expression of variational topology mass function $Q(\tau)$, they all rely on a subsplit Bayesian network (SBN) [41], which represents tree topology mass function $Q(\tau)$ as a product of conditional probabilities of splits (i.e., bipartition of the tip nodes) given their parent splits. However, SBN necessitates a preselection of the likely set of tree topologies, which hinders an end-to-end optimization strategy of the distribution over all the topologies and branch lengths.

---

[1]Our implementation is found at `https://github.com/m1m0r1/geophy`

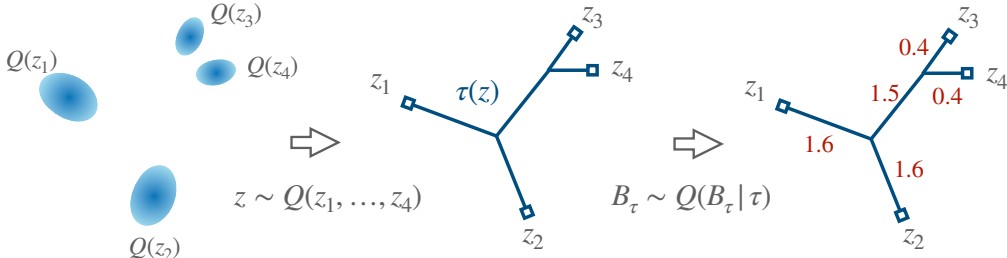

Figure 1: The proposed scheme for constructing a variational distribution $Q(\tau, B_\tau)$ of a tree topology and branch lengths by using a distribution $Q(z)$ defined on the continuous space. **Left**: The marginal distributions of four tip nodes $Q(z_1), \ldots, Q(z_4)$. **Middle**: Coordinates of the tip nodes $z = \{z_1, \ldots, z_4\}$ sampled from the distribution $Q(z)$, and a tree topology $\tau(z)$ determined from $z$. **Right**: A set of branch lengths $B_\tau$ (red figures) of the tree $\tau$ is sampled from a topology dependent distribution $Q(B_\tau|\tau)$.

### 2.3 Learnable topological features (LTFs)

Zhang [40] introduced a deterministic embedding method for tree topologies, termed learnable topological features (LTFs), which provide unique signatures for each tree topology. Specifically, given a set of nodes $V$ and edges $E$ of a tree topology $\tau$, a node embedding function $f_{\text{emb}} : V \to \mathbb{R}^d$ is determined to minimize the Dirichlet energy defined as follows:

$$\sum_{(u,v)\in E} \|f_{\text{emb}}(u) - f_{\text{emb}}(v)\|^2 . \tag{2}$$

When feature vectors for the tip nodes are determined, Zhang [40] showed that the optimal $f_{\text{emb}}$ for the interior nodes can be efficiently computed with two traversals of the tree topology. They also utilized $f_{\text{emb}}$ to parameterize the conditional distribution of branch length given tree topology $Q(B_\tau|\tau)$ by using graph neural networks with the input node features $\{f_{\text{emb}}(v)|v \in V\}$.

### 2.4 Hyperbolic spaces and probability distributions

Hyperbolic spaces are known to be able to embed hierarchical data with less distortion in considerably fewer dimensions than Euclidean spaces [32]. This property has been exploited to develop various data analysis applications such as link predictions for relational data [26], as well as phylogenetic analyses [21, 19]. For representing the uncertainty of embedded data on hyperbolic spaces, the wrapped normal distribution proposed in [25] has appealing characteristics: it is easy to sample from and evaluate the density of the distribution.

The Lorentz model $\mathbb{H}^d$ is a representation of the $d$-dimensional hyperbolic space, which is a sub-manifold of $d + 1$ dimensional Euclidean space. Denote $\mu \in \mathbb{H}^d$ and $\Sigma \in \mathbb{R}^{d \times d}$ be a location and scale parameters, respectively, a random variable $z \in \mathbb{H}^d$ that follows a wrapped normal distribution $\mathcal{WN}(\mu, \Sigma)$ is defined as follows:

$$u \sim \mathcal{N}(0, \Sigma), \quad z = \exp_\mu \circ \text{PT}_{\mu^o \to \mu}(u), \tag{3}$$

where $\mu^o$, $\exp_\mu : T_\mu \mathbb{H}^d \to \mathbb{H}^d$, and $\text{PT}_{\mu^o \to \mu} : T_{\mu^o} \mathbb{H}^d \to T_\mu \mathbb{H}^d$ denote the origin, the exponential map, and the parallel transport defined on the Lorentz model. Also, the probability density $\mathcal{WN}(\mu, \Sigma)$ has a closed form and is differentiable with respect to the parameters $\mu, \Sigma$. More details and properties are summarized in Appendix A.

## 3 Proposed methods

### 3.1 Geometric representations of tree topology ensembles

Considering the typically infeasible task of parameterizing the probability mass function of unrooted tree topologies $\mathcal{T}$, which requires $(2N - 5)!! - 1$ degrees of freedom, we propose an alternative approach. We suggest constructing the mass function $Q(\tau)$ through a transformation of a certain probability density $Q(z)$ over a continuous domain $\mathcal{Z}$, as follows:

$$Q(\tau) := \mathbb{E}_{Q(z)}[\mathbb{I}[\tau = \tau(z)]], \tag{4}$$

where $\tau : \mathcal{Z} \to \mathcal{T}$ denotes a deterministic link function that maps $N$ coordinates to the corresponding tree topology (Fig. 1). Note that we have overloaded $\tau$ to represent both a variable and function for notational simplicity. An example of the representation space $\mathcal{Z}$ is a product of the Euclidean space $\mathbb{R}^{N \times d}$ or hyperbolic space $\mathbb{H}^{N \times d}$, where $d$ denotes the dimension of each tip's representation coordinate. For the link function, we can use $\tau(z) = T_{\mathrm{NJ}} \circ D(z)$, where $D : \mathcal{Z} \to \mathbb{R}^{N \times N}$ denotes a function that takes $N$ coordinates and provides a distance matrix between those based on a geometric measure such as the Euclidean or hyperbolic distance. $T_{\mathrm{NJ}} : \mathbb{R}^{N \times N} \to \mathcal{T}$ denotes a map that takes this distance matrix and generates an unrooted binary tree topology of their phylogeny, determined using the Neighbor-Joining (NJ) algorithm [31]. While the NJ algorithm offers a rooted binary tree topology accompanied by estimated branch lengths, we only use the topology information and remove the root node from it to obtain the unrooted tree topology $\tau \in \mathcal{T}$.

### 3.2 Derivation of variational lower bound

Given a distribution of tip coordinates $Q(z)$ and an induced tree topology distribution $Q(\tau)$ according to equation (4), the variational lower bound (1) is evaluated as follows:

$$\mathcal{L}[Q] = \mathbb{E}_{Q(z)} \left[ \mathbb{E}_{Q(B_\tau | \tau(z))} \left[ \ln \frac{P(Y, B_\tau | \tau(z))}{Q(B_\tau | \tau(z))} \right] + \ln \frac{P(\tau(z))}{Q(\tau(z))} \right]. \tag{5}$$

Thanks to the deterministic mapping $\tau(z)$, we can obtain an unbiased estimator of $\mathcal{L}[Q]$ by sampling from $Q(z)$ without summing over the combinatorial many topologies $\mathcal{T}$. However, even when the density $Q(z)$ is computable, the evaluation of $\ln Q(\tau)$ remains still infeasible according to the definition (4). We resolve this issue by introducing the second lower bound with respect to a conditional variational distribution $R(z|\tau)$ as follows:

$$\mathcal{L}[Q, R] = \mathbb{E}_{Q(z)} \left[ \mathbb{E}_{Q(B_\tau | \tau(z))} \left[ \ln \frac{P(Y, B_\tau | \tau(z))}{Q(B_\tau | \tau(z))} \right] + \ln \frac{P(\tau(z)) R(z | \tau(z))}{Q(z)} \right]. \tag{6}$$

This formulation is similar in structure to the variational auto-encoders [14], where $Q(\tau|z)$ and $R(z|\tau)$ are viewed as a probabilistic decoder and encoder of data $\tau$, respectively. However, unlike typical auto-encoders, we design $Q(\tau|z)$ to be deterministic, and instead adjust $Q(z)$ to approximate $Q(\tau)$.

**Proposition 1.** Assuming that $\mathrm{supp}\, R(z|\tau) \supseteq \mathrm{supp}\, Q(z|\tau)$, the inequality $\ln P(Y) \geq \mathcal{L}[Q] \geq \mathcal{L}[Q, R]$ holds, where $\mathrm{supp}$ denotes the support of distribution. The first and the second equality holds when $Q(\tau, B_\tau) = P(\tau, B_\tau | Y)$ and $R(z|\tau) = Q(z|\tau)$, respectively.

*Proof.* The first variational lower bound of the marginal log-likelihood is given as follows:

$$\ln P(Y) \geq \ln P(Y) - D_{\mathrm{KL}} \left[ Q(\tau, B_\tau) \| P(\tau, B_\tau | Y) \right] = \mathbb{E}_{Q(\tau, B_\tau)} \left[ \ln \frac{P(Y, \tau, B_\tau)}{Q(\tau, B_\tau)} \right] := \mathcal{L}[Q], \tag{7}$$

where the equality condition of the first inequality holds when $Q(\tau, B_\tau) = P(\tau, B_\tau | Y)$. Since we have defined $Q(\tau)$ in equation (4), we can further transform the lower bound as $\mathcal{L}[Q]$

$$\mathcal{L}[Q] = \mathbb{E}_{Q(z)} \left[ \sum_{\tau \in \mathcal{T}} \mathbb{I}[\tau = \tau(z)] \, \mathbb{E}_{Q(B_\tau | \tau)} \left[ \ln \frac{P(Y, B_\tau | \tau)}{Q(B_\tau | \tau)} + \ln \frac{P(\tau)}{Q(\tau)} \right] \right], \tag{8}$$

from which equation (5) immediately follows. Hence the inequality $\ln P(Y) \geq \mathcal{L}[Q]$ and its equality condition $Q(\tau, B_\tau) = P(\tau, B_\tau | Y)$ have been proven.

Next, the entropy term $-\mathbb{E}_{Q(z)}[\ln Q(\tau(z))] = -\mathbb{E}_{Q(\tau)}[\ln Q(\tau)]$ in equation (5) can be transformed to derive further lower bound as follows:

$$-\mathbb{E}_{Q(\tau)}[\ln Q(\tau)] \geq -\mathbb{E}_{Q(\tau)}[\ln Q(\tau) + D_{\mathrm{KL}}(Q(z|\tau)\|R(z|\tau))]$$

$$= -\mathbb{E}_{Q(\tau)Q(z|\tau)}\left[\ln \frac{Q(\tau)Q(z|\tau)}{R(z|\tau)}\right] = -\mathbb{E}_{Q(z)}\left[\ln \frac{Q(z)}{R(z|\tau(z))}\right],$$

where the last equality is derived by using the relation $\mathbb{E}_{Q(\tau)Q(z|\tau)}[\cdot] = \mathbb{E}_{Q(z)}[\sum_\tau \mathbb{I}[\tau = \tau(z)]\cdot]$ and $Q(\tau)Q(z|\tau) = Q(z)\mathbb{I}[\tau = \tau(z)]$. The equality condition of the first inequality holds when $R(z|\tau) = Q(z|\tau)$. Hence, the inequality $\mathcal{L}[Q] \geq \mathcal{L}[Q, R]$ and the equality condition is proven.  $\square$

Similar to Burda et al. [1], we can also derive a tractable importance-weighted lower-bound of the model evidence (IW-ELBO), which is used for estimating the marginal log-likelihood (MLL), $\ln P(Y)$, or an alternative lower-bound objective for maximization. The details and derivations are described in Appendix B.

### 3.3 Differentiable phylogenetic inference with GeoPhy

---
**Algorithm 1** GeoPhy algorithm

---
1: $\theta, \phi, \psi \leftarrow$ Initialize variational parameters
2: **while** not converged **do**
3: $\quad \epsilon_z^{(1:K)}, \epsilon_B^{(1:K)} \leftarrow$ Random samples from distributions $p_z, p_B$
4: $\quad z^{(1:K)} \leftarrow h_\theta(\epsilon_z^{(1:K)})$
5: $\quad B_\tau^{(1:K)} \leftarrow h_\phi(\epsilon_B^{(1:K)}, \tau(z^{(1:K)}))$
6: $\quad \widehat{g}_\theta, \widehat{g}_\phi, \widehat{g}_\psi \leftarrow$ Estimate the gradients $\nabla_\theta \mathcal{L}, \nabla_\phi \mathcal{L}, \nabla_\psi \mathcal{L}$
7: $\quad \theta, \phi, \psi \leftarrow$ Update parameters using an SGA algorithm, given gradients $\widehat{g}_\theta, \widehat{g}_\phi, \widehat{g}_\psi$
8: **end while**
9: **return** $\theta, \phi, \psi$

---

Here, we introduce parameterized distributions, $Q_\theta(z)$, $Q_\phi(B_\tau|\tau)$, and $R_\psi(z|\tau)$, aiming to optimize the variational objective $\mathcal{L}[Q_{\theta,\phi}, R_\psi]$ using stochastic gradient ascent (SGA). The framework, which we term GeoPhy, is summarized in Algorithm 1. While aligning with black-box variational inference [27] and its extension to phylogenetic inference [42], our unique lower bound objective enables us to optimize the distribution over entire phylogenetic trees within continuous geometric spaces.

**Gradients of lower bound**   We derive the gradient terms for stochastic optimization of $\mathcal{L}$, omitting parameters $\theta, \phi, \psi$ for notational simplicity. We assume that $Q_\theta(z)$ and $Q_\phi(B_\tau|\tau(z))$ are both reparameterizable; namely, we can sample from these distributions as follows:

$$z = h_\theta(\epsilon_z), \quad B_\tau = h_\phi(\epsilon_B, \tau(z)), \tag{9}$$

where the terms $\epsilon_z \sim p_z(\epsilon_z)$ and $\epsilon_B \sim p_B(\epsilon_B)$ represent sampling from parameter-free distributions $p_z$ and $p_B$, respectively, and the functions $h_\theta$ and $h_\phi$ are differentiable with respect to $\theta$ and $\phi$, respectively. Although we cannot take the derivative of $\tau(z)$ with respect to $z$, the gradient of $\mathcal{L}$ with respect to $\theta$ is evaluated as follows:

$$\nabla_\theta \mathcal{L} = \mathbb{E}_{Q_\theta(z)}\left[(\nabla_\theta \ln Q_\theta(z))\left(\mathbb{E}_{Q_\phi(B_\tau|\tau(z))}\left[\ln \frac{P(Y, B_\tau|\tau(z))}{Q_\phi(B_\tau|\tau(z))}\right] + \ln P(\tau(z))R_\psi(z|\tau(z))\right)\right]$$
$$+ \nabla_\theta \mathbb{H}[Q_\theta(z)], \tag{10}$$

where $\mathbb{H}$ denotes the differential entropy. The gradient of $\mathcal{L}$ with respect to $\phi$ and $\psi$ are evaluated as follows:

$$\nabla_\phi \mathcal{L} = \mathbb{E}_{Q_\theta(z)} \mathbb{E}_{p_B(\epsilon)}\left[\nabla_\phi \ln \frac{P(Y, B_\tau = h_\phi(\epsilon, \tau)|\tau(z))}{Q_\phi(B_\tau = h_\phi(\epsilon, \tau)|\tau(z))}\right], \quad \nabla_\psi \mathcal{L} = \mathbb{E}_{Q_\theta(z)}\left[\nabla_\psi \ln R_\psi(z|\tau(z))\right], \tag{11}$$

where we assume a tractable density model for $R_\psi(z|\tau)$.

**Gradient estimators and variance reduction**  From equation (10), an unbiased estimator of $\nabla_\theta \mathcal{L}$, using $K$-sample Monte Carlo samples, can be derived as follows:

$$\widehat{g}_\theta^{(K)} = \frac{1}{K} \sum_{k=1}^{K} \left( \nabla_\theta \ln Q_\theta(z^{(k)}) \cdot f(z^{(k)}, B_\tau^{(k)}) - \nabla_\theta \ln Q_\theta(h_\theta(\epsilon_z^{(k)})) \right), \qquad (12)$$

where we denote $\epsilon_z^{(k)} \sim p_z$ $z^{(k)} = h_\theta(\epsilon_z^{(k)})$, $\epsilon_B^{(k)} \sim p_B$, $B_\tau^{(k)} = h_\phi(\epsilon_B^{(k)}, \tau(z^{(k)}))$, and

$$f(z, B_\tau) := \ln \frac{P(Y, B_\tau | \tau(z))}{Q_\phi(B_\tau | \tau(z))} + \ln P(\tau(z)) R_\psi(z | \tau(z)). \qquad (13)$$

Note that we explicitly distinguish $z$ and $h_\theta(\epsilon_z)$ to indicate the target of differentiation with respect to $\theta$.

In practice, it is known to be crucial to reducing the variance of the gradient estimators proportional to the score function $\nabla_\theta \ln Q_\theta$ to make optimization feasible. This issue is addressed by introducing a control variate $c(z)$, which has a zero expectation $\mathbb{E}_z[c(z)] = 0$ and works to reduce the variance of the term by subtracting it from the gradient estimator. For the case of $K > 1$, it is known to be effective to use simple Leave-one-out (LOO) control variates [15, 28] as follows:

$$\widehat{g}_{\theta,\text{LOO}}^{(K)} = \frac{1}{K} \sum_{k=1}^{K} \left[ \nabla_\theta \ln Q_\theta(z^{(k)}) \cdot \left( f(z^{(k)}, B_\tau^{(k)}) - \overline{f_k}(z^{(\backslash k)}, B_\tau^{(\backslash k)}) \right) - \nabla_\theta \ln Q_\theta(h_\theta(\epsilon_z^{(k)})) \right], \tag{14}$$

where we defined $\overline{f_k}(z^{(\backslash k)}, B_\tau^{(\backslash k)}) := \frac{1}{K-1} \sum_{k'=1, k' \neq k}^{K} f(z^{(k)}, B_\tau^{(k)})$, which is a constant with respect to the $k$-the sample.

We can also adopt a gradient estimator called LAX [7], which uses a learnable surrogate function $s_\chi(z)$ differentiable in $z$ to form control variates as follows:

$$\widehat{g}_{\theta,\text{LAX}} = (\nabla_\theta \ln Q_\theta(z)) \left( f(z, B_\tau) - s_\chi(z) \right) + \nabla_\theta s_\chi(h_\theta(\epsilon_z)) - \nabla_\theta \ln Q_\theta(h_\theta(\epsilon_z)), \qquad (15)$$

where we write the case with $K = 1$ for simplicity. To optimize the surrogate function $s_\chi$, we use the estimator $\widehat{g}_\chi = \nabla_\chi \left\langle \widehat{g}_\theta^2 \right\rangle$ [2], where $\langle \cdot \rangle$ denotes the average of the vector elements. We summarize the procedure with LAX estimator in Algorithm 2.

We also consider other options for the gradient estimators, such as the combinations of the LOO and LAX estimators, the gradient of IW-ELBO, and its variance-reduced estimator called VIMCO [22] in our experiments. For the remaining gradients terms $\nabla_\phi \mathcal{L}$ and $\nabla_\psi \mathcal{L}$, we can simply employ reparameterized estimators as follows:

$$\widehat{g}_\phi = \nabla_\phi \ln \frac{P(Y, B_\tau = h_\phi(\epsilon_B, \tau) | \tau(z))}{Q_\phi(B_\tau = h_\phi(\epsilon_B, \tau) | \tau(z))}, \quad \widehat{g}_\psi = \nabla_\psi \ln R_\psi(z | \tau(z)), \qquad (16)$$

where we denote $\epsilon_B \sim p_B$ and $z \sim Q_\theta$. More details of the gradient estimators are summarized in Appendix B.

**Variational distributions**  To investigate the basic effectiveness of GeoPhy algorithm, we employ simple constructions for the variational distributions $Q_\theta(z)$, $Q_\phi(B_\tau | \tau)$, and $R_\psi(z | \tau)$. We use an independent distribution for each tip node coordinate, i.e. $Q_\theta(z) = \prod_{i=1}^{N} Q_{\theta_i}(z_i)$, where we use a $d$-dimensional normal or wrapped normal distribution for the coordinates of each tip node $z_i$. For the conditional distribution of branch lengths given tree topology, $Q_\phi(B_\tau | \tau)$, we use the diagonal lognormal distribution whose location and scale parameters are given as a function of the LTFs of the topology $\tau$, which is comprised of GNNs as proposed in [40]. For the model of $R_\psi(z | \tau)$, we also employ an independent distribution: $R_\psi(z | \tau) = \prod_{i=1}^{N} R_{\psi_i}(z_i | \tau)$, where, we use the same type of distribution as $Q_{\theta_i}(z_i)$, independent of $\tau$.

---

[2] $\widehat{g}_\chi$ is an unbiased estimator of $\nabla \left\langle \mathbb{V}_{Q(z)}[(\widehat{g}_\theta)] \right\rangle$ as shown in [7].

# 4 Related work

**Differentiability for discrete optimization** Discrete optimization problems often suffer from the lack of informative gradients of the objective functions. To address this issue, continuous relaxation for discrete optimization has been actively studied, such as a widely-used reparameterization trick with the Gumbel-softmax distribution [10, 20]. Beyond categorical variables, recent approaches have further advanced the continuous relaxation techniques to more complex discrete objects, including spanning trees [33]. However, it is still nontrivial to extend such techniques to the case of binary tree topologies. As outlined in equation (4), we have introduced a distribution over binary tree topologies $\mathcal{T}$ derived from continuous distributions $Q(z)$. This method facilitates a gradient-based optimization further aided by variance reduction techniques.

**Gradient-based algorithms for tree optimization** For the hierarchical clustering (HC), which reconstructs a tree relationship based on the distance measures between samples, gradient-based algorithms [23, 2, 3] have been proposed based on Dasgupta's cost function [4]. In particular, Chami et al. [2] proposed to decode tree topology from hyperbolic coordinates while the optimization is performed for a relaxed cost function, which is differentiable with respect to the coordinates. However, these approaches are not readily applicable to more general problems, including phylogenetic inference, as their formulations depend on the specific form of the cost functions.

**Phylogenetic analysis in hyperbolic space** The approach of embedding phylogenetic trees into hyperbolic spaces has been explored for visualization and an interpretation of novel samples with existing phylogeny [21, 11]. For the inference task, a maximum-likelihood approach was proposed in [35], which however assumed a simplified likelihood function of pairwise distances. A recent study [19] proposed an MCMC-based algorithm for sampling $(\tau, B_\tau)$, which were linked from coordinates $z$ using the NJ algorithm [31]. However, there remained the issue of an unevaluated Jacobian determinant, which posed a challenge in evaluating inference objectives. Given that we only use topology $\tau$ as described in equation (4), the variational lower bound for the inference can be unbiasedly evaluated through sampling, as shown in Proposition 1.

# 5 Experiments

We applied GeoPhy to perform phylogenetic inference on biological sequence datasets of 27 to 64 species compiled in Lakner et al. [17].

**Models and training** As a prior distribution of $P(\tau)$ and $P(B_\tau|\tau)$, we assumed a uniform distribution over all topologies, and an exponential distribution $\text{Exp}(10)$ independent for all branches, respectively, as commonly used in the literature [42, 16]. For the neural network used in the parameterization of $Q_\phi(B_\tau|\tau)$, we employed edge convolutional operation (EDGE), which was well-performed architecture in [40]. For the stochastic gradient optimizations, we used the Adam optimizer [13] with a learning rate of 0.001. We trained for GeoPhy until one million Monte Carlo tree samples were consumed for the gradient estimation of our loss function. This number equals the number of likelihood evaluations (NLEs) and is used for a standardized comparison of experimental runs [34, 42]. The marginal log-likelihood (MLL) value is estimated with 1000 MC samples. More details of the experimental setup are found in Appendix C.

**Initialization of coordinates** We initialized the mean parameters of the tip coordinate distribution $Q_\theta(z)$ with the multi-dimensional scaling (MDS) algorithm when $Q_\theta$ was given as normal distributions. For $Q_\theta$ comprised of wrapped normal distributions, we used the hyperbolic MDS algorithm (hMDS) proposed in [32] for the initialization. For a distance matrix used for MDS and hMDS, we used the Hamming distance between each pair of the input sequences $Y$ as similar to [19]. For the scale parameters, we used 0.1 for all experiments. For $R_\psi(z)$, we used the same mean parameters as $Q_\theta(z)$ and 1.0 for the scale parameters.

## 5.1 Exploration of stable learning conditions

To achieve a stable and fast convergence in the posterior approximations, we compared several control variates (CVs) for the variance reduction of the gradients by using DS1 dataset [8]. We

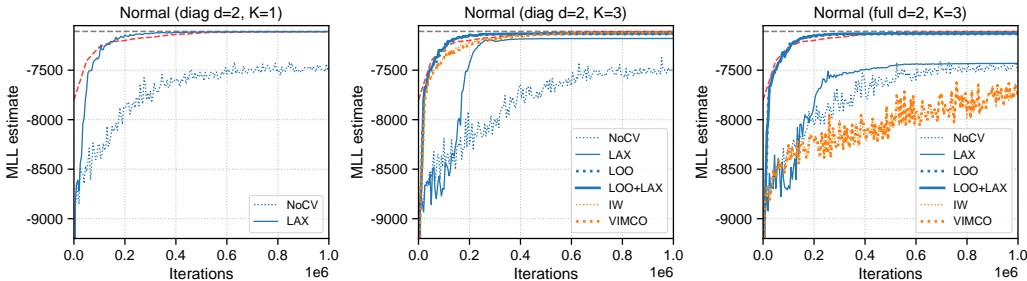

Figure 2: Comparison of marginal log-likelihood (MLL) estimates for DS1 dataset with different control variates. For a variational distribution, $Q(z)$, an independent two-dimensional Normal distribution with a diagonal (diag) or full covariance matrix was used for each tip node. $K$ stands for the number of Monte Carlo samples used for gradient estimation. For reference, we show the mean MLL value (gray dashed lines; $-7108.42 \pm 0.18$) estimated with the MrBayes stepping-stone (SS) method in [42]. We also show MLL estimates obtained with VBPI-GNN (red dashed lines) using VIMCO estimator ($K = 10$). The iterations are counted as the number of likelihood evaluations (NLEs). Legend: NoCV stands for no control variates, LOO for leave-one-out CV, and IW for the case of using importance-weighted ELBO as the learning objective.

demonstrate that the choice of control variates is crucial for optimization (Fig. 2). In particular, adaptive control variates (LAX) for individual Monte Carlo samples and the leave-one-out (LOO) CVs for multiple Monte Carlo samples (with $K = 3$) significantly accelerate the convergence of the marginal log-likelihood (MLL) estimates, yielding promising results comparable to VBPI-GNN. Although IW-ELBO was effective when $Q(z)$ was comprised of diagonal Normal distributions, no advantages were observed for the case with full covariance matrices. Similar tendencies in the MLL estimates were observed for the other dimension ($d = 3, 4$) and wrapped normal distributions (Appendix D.1).

## 5.2 Comparison of different topological distributions

We investigated effective choices of tip coordinate distributions $Q(z)$, which yielded tree topologies, by comparing different combinations of distribution types (normal $\mathcal{N}$ or wrapped normal $\mathcal{WN}$), space dimensions ($d = 2, 3, 4$), and covariance matrix types (diagonal or full), across selected CVs (LAX, LOO, and LOO+LAX). While overall performance was stable and comparable for the range of configurations, the most flexible ones: the normal and wrapped normal distributions with $d = 4$ and a full covariance matrix, indicated relatively high MLL estimates within their respective groups (Table 2 in Appendix D), implying the importance of flexibility in the variational distributions.

## 5.3 Performance evaluation across eight benchmark datasets

To demonstrate the inference performance of GeoPhy, we compared the marginal log-likelihood (MLL) estimates for the eight real datasets (DS1-8) [8, 6, 37, 9, 17, 43, 38, 30]. The gold-standard values were obtained using the stepping-stone (SS) algorithm [36] in MrBayes [29], wherein each evaluation involved ten independent runs, each with four chains of 10,000,000 iterations, as reported in [42]. In Table 1, we summarize the MLL estimates of GeoPhy and other approximate Bayesian inference approaches for the eight datasets. While VBPI-GNN [40] employs a preselected set of tree topologies as its support set before execution, it is known to provide reasonable MLL estimates near the reference values. The other approaches including CSMC [34], VCSMC [24], $\phi$-CSMC [16] and GeoPhy (ours) tackles the more challenging general problem of model optimization by considering all candidate topologies without preselection. We compared the GeoPhy for two configurations of $Q(z)$: a wrapped normal distribution $\mathcal{WN}$ with a 4-dimensional full covariance matrix, and a 2-dimensional diagonal matrix, with three choices of CVs for optimization. Results for other settings, including those of the normal distributions in Euclidean spaces, are provided in Table 3 in Appendix D. There, we noted a slight advantage of wrapped normal distributions over their Euclidean counterparts. In

Table 1, we observe that the superior performance of the larger model $\mathcal{WN}$(full,4) over $\mathcal{WN}$(diag,2), observed in Table 2, is consistently reproduced across all datasets. Moreover, although $\mathcal{WN}$(full,2) is relatively less performant in our comparison, almost all the estimates outperformed the other CSMC-based methods, implying the stability and efficiency of our approach.

Table 1: Comparison of the marginal log-likelihood (MLL) estimates with different approaches in eight benchmark datasets. The MLL values for MrBayes SS and VBPI-GNN, which employs the VIMCO estimator of 10 samples and EDGE GNN, are sourced from [40], The values of CSMC, VCSMC, and $\phi$-CSMSC are referenced from [16]. We adopt the nomenclature from the literature [42] and refer to the dataset named DS7 in [16] as DS8. The MLL values for our approach (GeoPhy) are shown for two different $Q(z)$ configurations: a wrapped normal distribution $\mathcal{WN}$ with 4-dimensional full covariance matrix, and 2-dimensional diagonal matrix. In each training, we used three different sets of CVs: LAX with $K = 1$, LOO with $K = 3$ which we labeled LOO(3), and a combination of LOO and LAX, denoted as LOO(3)+. The bold figures are the best (highest) values obtained with GeoPhy and the tree CSMC-based methods, all of which perform an approximate Bayesian inference without the preselection of topologies. We underlined GeoPhy's MLL estimates that outperformed the other CSMC-based methods, demonstrating superior performance across various configurations.

| Dataset | DS1 | DS2 | DS3 | DS4 | DS5 | DS6 | DS7 | DS8 |
|---|---|---|---|---|---|---|---|---|
| #Taxa ($N$) | 27 | 29 | 36 | 41 | 50 | 50 | 59 | 64 |
| #Sites ($M$) | 1949 | 2520 | 1812 | 1137 | 378 | 1133 | 1824 | 1008 |
| MrBayes SS | $-7108.42$ | $-26367.57$ | $-33735.44$ | $-13330.06$ | $-8214.51$ | $-6724.07$ | $-37332.76$ | $-8649.88$ |
| [29, 42] | (0.18) | (0.48) | (0.5) | (0.54) | (0.28) | (0.86) | (2.42) | (1.75) |
| VBPI-GNN | $-7108.41$ | $-26367.73$ | $-33735.12$ | $-13329.94$ | $-8214.64$ | $-6724.37$ | $-37332.04$ | $-8650.65$ |
| [40] | (0.14) | (0.07) | (0.09) | (0.19) | (0.38) | (0.4) | (0.26) | (0.45) |
| CSMC | $-8306.76$ | $-27884.37$ | $-35381.01$ | $-15019.21$ | $-8940.62$ | $-8029.51$ | — | $-11013.57$ |
| [34, 16] | (166.27) | (226.6) | (218.18) | (100.61) | (46.44) | (83.67) | — | (113.49) |
| VCSMC | $-9180.34$ | $-28700.7$ | $-37211.2$ | $-17106.1$ | $-9449.65$ | $-9296.66$ | — | — |
| [24, 16] | (170.27) | (4892.67) | (397.97) | (362.74) | (2578.58) | (2046.7) | — | — |
| $\phi$-CSMC | $-7290.36$ | $-30568.49$ | $-33798.06$ | $-13582.24$ | $-8367.51$ | $-7013.83$ | — | $-9209.18$ |
| [16] | (7.23) | (31.34) | (6.62) | (35.08) | (8.87) | (16.99) | — | (18.03) |
| $\mathcal{WN}$(full,4) | __$-7111.55$__ | $-26379.48$ | $-33757.79$ | $-13342.71$ | $-8240.87$ | $-6735.14$ | $-37377.86$ | $-8663.51$ |
| | (0.07) | (11.60) | (8.07) | (1.61) | (9.80) | (2.64) | (29.48) | (6.85) |
| LOO(3) | $-7119.77$ | __$-26368.44$__ | $-33736.01$ | $-13339.26$ | $-8234.06$ | __$-6733.91$__ | __$-37350.77$__ | $-8671.32$ |
| | (11.80) | (0.13) | (0.03) | (3.19) | (0.57) | (0.57) | (11.74) | (5.99) |
| LOO(3)+ | $-7116.09$ | $-26368.54$ | __$-33735.85$__ | __$-13337.42$__ | $-8233.89$ | $-6735.90$ | $-37358.96$ | __$-8660.48$__ |
| | (10.67) | (0.12) | (0.12) | (1.32) | (6.63) | (1.13) | (13.06) | (0.78) |
| $\mathcal{WN}$(diag,2) | $-7126.89$ | $-26444.84$ | $-33823.74$ | $-13358.16$ | $-8251.45$ | $-6745.60$ | $-37516.88$ | $-8719.44$ |
| | (10.06) | (27.91) | (15.62) | (9.79) | (9.72) | (8.36) | (69.88) | (60.54) |
| LOO(3) | $-7130.67$ | $-26380.41$ | $-33737.75$ | $-13346.94$ | $-8239.36$ | $-6741.63$ | $-37382.28$ | $-8690.41$ |
| | (10.67) | (14.40) | (2.48) | (4.25) | (4.62) | (3.23) | (31.96) | (15.92) |
| LOO(3)+ | $-7128.40$ | $-26375.28$ | $-33736.91$ | $-13347.32$ | $-8235.41$ | $-6742.40$ | $-37411.28$ | $-8683.22$ |
| | (9.78) | (11.78) | (1.91) | (4.42) | (5.70) | (1.94) | (56.74) | (13.13) |

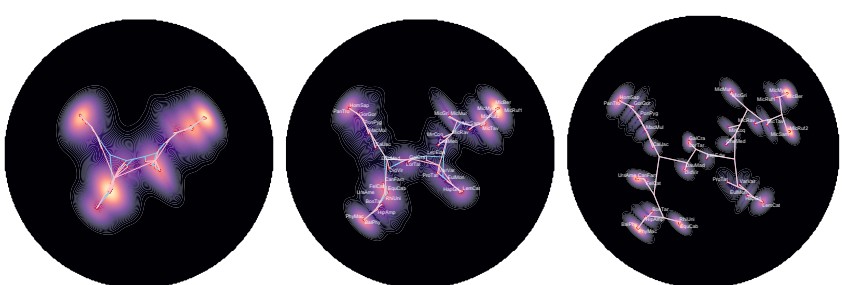

Figure 3: Superimposed probability densities of topological distributions $\sum_{i=1}^{N} Q(z_i)$ up to 100,000, 200,000, and 500,000 steps (MC samples) from the left. For $Q(z)$, we employed a wrapped normal distribution with a two-dimensional full covariance matrix. The experiments used the DS3 dataset ($N = 36$). The majority-rule consensus phylogenetic tree obtained with MrBayes and each step of GeoPhy are shown in blue and red lines, respectively. The center area is magnified by transforming the radius $r$ of the Poincaré coordinates into $\tanh 2.1r$.

We further analyze the accuracy and expressivity of tree topology distribution $Q(\tau)$ in comparison with MCMC tree samples. As the training progresses, the majority consensus tree from GeoPhy

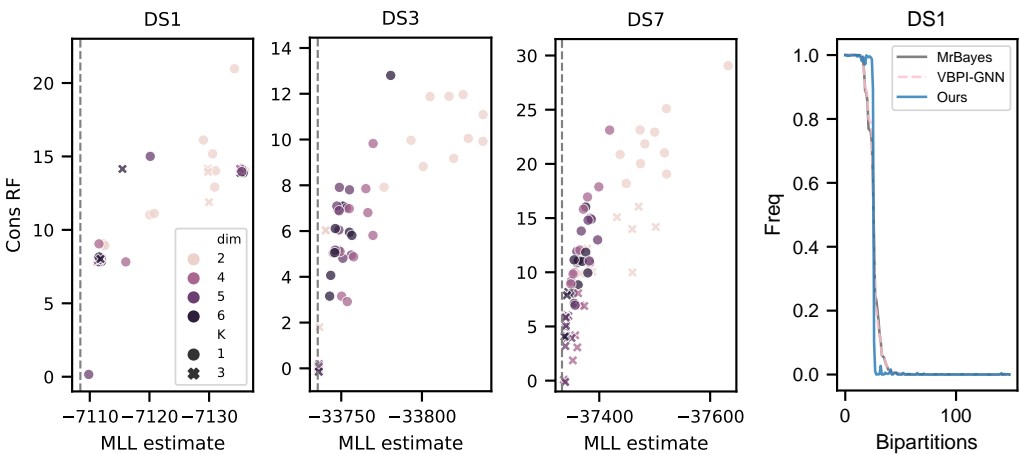

Figure 4: **First** to **Third**: Marginal log-likelihood (MLL) estimates and Robinson-Foulds (RF) distance between the consensus trees obtained from topology distribution $Q(\tau)$ and MrBayes (MCMC) for datasets DS1, DS3, and DS7, respectively. **Fourth**: Comparison of bipartition frequencies derived from the posterior distribution of tree topologies for MrBayes, VBPI-GNN, and GeoPhy (ours).

progressively aligns with those obtained via MCMC (Fig. 3). Furthermore, we also observed that the Robinson-Foulds (RF) distance between the consensus trees derived from the topological distribution $Q(\tau)$ and MCMC (MrBayes) are highly aligned with the MLL estimates (Fig. 4 First to Third), underscoring the validity of MLL-based evaluations. While GeoPhy provides a tree topology mode akin to MCMCs with a close MLL value, it may require more expressivity in $Q(z)$ to fully represent tree topology distributions. In Fig. 4 Fourth, we illustrate the room for improvement in the expressivity of $Q(\tau)$ in representing the bipartition frequency of tree samples, where VBPI-GNN aligns more closely with MrBayes. Further analysis is found in Appendix D.3.

In terms of execution time, we compared GeoPhy with VBPI-GNN and MrBayes, obtaining results that are promising and comparable to VBPI-GNN (Fig. 10 Left). Though we have not included CSMC-based methods in the comparison due to deviations in MLL values, they tend to capitalize on parallelism for faster execution as demonstrated in [16].

## 6  Conclusion

We developed a novel differential phylogenetic inference framework named GeoPhy, which optimized a variational distribution of tree topology and branch lengths without the preselection of candidate topologies. We also proposed a practical implementation for stable model optimization through choices of distribution models and control variates. In experiments conducted with real sequence datasets, GeoPhy consistently outperformed other approximate Bayesian methods that considered whole topologies.

## 7  Limitations and Future work

Although GeoPhy exhibits remarkable performance on standard benchmarks without preselecting topologies, it may be necessary to use more expressive distributions for $Q(z)$ than independent parametric distributions to reach a level of performance comparable to state-of-the-art VBPI or gold-standard MCMC evaluations. Another limitation of our study lies in its exclusive focus on efficiency, measured in terms of the number of likelihood evaluations (NLEs), without paying significant attention to optimizing the computational cost per iteration. The general design of our variational distribution allows us to replace the tree link function such as UPGMA as presented in Fig. 9 Right. Future work should explore alternative design choices in continuous tree topology representations and the link functions to address these limitations. Extending our framework to include complex models remains crucial. Given the potential of scalable phylogenetic methods to enhance our understanding of viral and bacterial evolution, further exploration could substantially impact public health and disease control.

## Acknowledgments and Disclosure of Funding

We appreciate the valuable comments on our study provided by Dr. Tsukasa Fukunaga of Waseda University and the anonymous reviewers. TM was partially supported by JSPS KAKENHI JP20H04239 and JST ACT-X JPMJAX22AI. MH was partially supported by JST CREST JPMJCR21F1 and AMED JP22ama121055, JP21ae0121049 and JP21gm0010008.

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

# A  Details of Background

## A.1  Summary of calculations for wrapped normal distributions

The wrapped normal distribution, as proposed in [25], is a probability distribution defined on hyperbolic spaces, which is easy to sample from and evaluate its probability density at arbitrary coordinates in the hyperbolic spaces. In the following, we provide a detailed summary of the calculation involved in applying the wrapped normal distributions within the context of the Lorentz model of hyperbolic spaces.

The Lorentz model, denoted as $\mathbb{H}^d$, represents the $d$-dimenstional hyperbolic space as a submanifold of a $d + 1$ dimensional Euclidean space. Given $u, v \in \mathbb{R}^{d+1}$, we can define the pseudo-inner product and pseudo-norm as follows:

$$\langle u, v \rangle_L := -u_0 v_0 + \sum_{j=1}^{d} u_j v_j, \quad \|u\|_L := \sqrt{\langle u, u \rangle_L}. \tag{17}$$

The distance between hyperbolic coordinates $\nu, \mu \in \mathbb{H}^d$ is defined as follows:

$$\mathrm{d}(\nu, \mu) := \cosh^{-1}(-\langle \nu, \mu \rangle_L), \tag{18}$$

where $\cosh^{-1}$ denotes the inverse function of the hyperbolic cosine function. Consider hyperbolic coordinates $\nu, \mu \in \mathbb{H}^d$ and tangent vectors $u \in T_\mu \mathbb{H}^d$ and $v \in T_\nu \mathbb{H}^d$. An exponential map $\exp_\mu(u) \in \mathbb{H}^d$, a logarithm map $\log_\mu(\nu) \in T_\mu \mathbb{H}^d$, and a parallel transport map $\mathrm{PT}_{\nu \to \mu}(v) \in T_\mu \mathbb{H}^d$, can be calculated as follows:

$$\exp_\mu(u) = \cosh(\|u\|_L)\mu + \sinh(\|u\|_L)\frac{u}{\|u\|_L}, \tag{19}$$

$$\log_\mu(\nu) = \frac{\cosh^{-1}(\alpha)}{\sqrt{\alpha^2 - 1}} \left( \nu - \alpha\mu \right), \tag{20}$$

$$\mathrm{PT}_{\nu \to \mu}(v) = v + \frac{\langle \mu - \alpha\nu, v \rangle_L}{\alpha + 1}(\nu + \mu), \tag{21}$$

where we denote $\alpha = -\langle \nu, \mu \rangle_L$, and $\cosh^{-1}$ represents the inverse function of $\cosh$.

Given location and scale parameters denoted as $\mu \in \mathbb{H}^d$ and $\Sigma \in \mathbb{R}^{d \times d}$, respectively, the procedure for sampling from a wrapped normal distribution $z \sim \mathcal{WN}(\mu, \Sigma)$ defined over $\mathbb{H}^d$ is given as follows:

$$z = \exp_\mu \circ \mathrm{PT}_{\mu^o \to \mu}(u), \quad u_{1:d} \sim \mathcal{N}(0, \Sigma), \tag{22}$$

where $\mu^o = (1, 0, \ldots, 0)^\top$ denotes the origin of the $\mathbb{H}^d$. Note that we set $u_0 = 0$, and $u := u_{0:d} \in T_{\mu^o} \mathbb{H}^d$ represents a tangent vector at the origin $T_{\mu^o} \mathbb{H}^d$.

From the sampling definition in equation (22), the probability density function $\mathcal{WN}(z; \mu, \Sigma)$ can be derived as follows:

$$\log \mathcal{WN}(z; \mu, \Sigma) = \log \mathcal{N}(u; 0, \Sigma) - (d - 1) \ln \left( \frac{\sinh \|u\|_L}{\|u\|_L} \right), \tag{23}$$

where $u$ is defined as $u = \mathrm{PT}_{\mu \to \mu^o} \circ \log_\mu(z)$. For detailed derivation, we refer to Appendix A of [25].

## A.2  GNN-based parameterization for variational branch length distributions

In this work, we employ a variational branch length distribution $Q_\phi(B_\tau | \tau)$ parameterized with a graph neural network (GNN) as described in [40]. In concrete, each of the branch lengths follows an independent lognormal distribution, where its location and scale parameters are predicted with a GNN that takes the tree topology $\tau$ and the learnable topological features (LTFs) of the topology $\tau$, which are computed with a method described in [40]. Below, we summarize an architecture that we use in this study.

**Branch length parameterizations**   Let $V_\tau$ and $E_\tau$ respectively represent the sets of nodes and branch edges for a given unrooted binary tree topology $\tau$. The input to the GNN consists of node features represented by LTFs denoted as $\{h_v^{(0)}\}_{v \in V_\tau}$. These features undergo transformation $L$ times as follows:

$$\{h_v^{(L)}\}_{v \in V_\tau} = \text{GNN}(\{h_v^{(0)}\}_{v \in V_\tau}) = g^{(L)} \circ \cdots \circ g^{(1)}(\{h^{(0)}\}_{v \in V_\tau}), \tag{24}$$

where we set $L = 2$. The function $g^{(\ell)}$ represents a GNN layer. For this function, we utilize edge convolutional layers, which will be described in more detail in the following paragraph.

Next, the last node features $h^{(L)}$ are transformed to output parameters of edge length as follows:

$$\widetilde{h}_v = \text{MLP}_V(h_v^{(L)}), \qquad (\forall v \in V_\tau) \tag{25}$$

$$\widetilde{m}_{(v,u)} = \text{MAX}(\widetilde{h}_v, \widetilde{h}_u), \qquad (\forall (v,u) \in E_\tau) \tag{26}$$

$$\mu_{(v,u)}, \log \sigma_{(v,u)} = \text{MLP}_E(\widetilde{m}_{(v,u)}) \qquad (\forall (v,u) \in E_\tau) \tag{27}$$

where $\text{MLP}_N$, $\text{MAX}$, and $\text{MLP}_E$ denotes a multi-layer perceptron for node features with two hidden layers, the element-wise max operation, and a multi-layer perceptron with a hidden layer that outputs the location and scale parameter $(\mu_e, \sigma_e)$ of the lognormal distributions for each edge $e \in \mathbb{E}_\tau$. For each of the hidden layers employed in $\text{MLP}_N$ and $\text{MLP}_E$, we set its width to 100 and apply the ELU activation function after the linear transformation of input values.

**Edge convolutional layers**   In a previous study [40], a GNN with edge convolutional layers, referred to as EDGE, demonstrated strong performance when predicting the posterior tree distributions. In EDGE, the function $g^{(\ell)}$ transforms node features $\{h_v^{(\ell)}\}_{v \in V_\tau}$ according to the following scheme:

$$\{h_v^{(\ell+1)}\}_{v \in V_\tau} = g^{(\ell)}(\{h_v^{(\ell)}\}_{v \in V_\tau}), \tag{28}$$

where $g^{(\ell)}$ is comprised of the edge convolutional operation with the exponential linear unit (ELU) activation function. Specifically, the transformation with the layer $g^{(\ell)}$ is computed as follows:

$$e_{u \to v}^{(\ell)} = \text{MLP}^{(\ell)}\left(h_v^{(\ell)} \| h_u^{(\ell)} - h_v^{(\ell)}\right), \quad \forall u \in N_\tau(v) \tag{29}$$

$$h_v'^{(\ell+1)} = \underset{u \in N_\tau(v)}{\text{AGG}^{(\ell)}} e_{u \to v}^{(\ell)}, \tag{30}$$

$$h_v^{(\ell+1)} = \text{ELU}\left(h_v'^{(\ell+1)}\right), \tag{31}$$

where $N_\tau(v)$ represents a set of neighboring nodes connected to node $v$ in the tree topology $\tau$, $\|$ refers to the concatenation operation of elements, $\text{MLP}^{(\ell)}$ denotes a full connection layer and the exponential linear unit (ELU) activation unit, and $\text{AGG}^{(\ell)}$ represents an aggregation operation that takes the maximum value of neighboring edge features $e_{u \to v}^{(\ell)}, \forall u \in N(v)$ for each element.

# B   Variational Lower Bounds and Gradient Estimators

## B.1   Variational lower bound

In Proposition 1, we present that the following functional is a lower bound of the marginal log-likelihood $\ln P(Y)$.

$$\mathcal{L}[Q, R] := \mathbb{E}_{Q(z, B_\tau)}\left[\ln F'(z, B_\tau)\right] = \mathbb{E}_{Q(z)}\left[\mathbb{E}_{Q(B_\tau|z)}[\ln F(z, B_\tau)] - \ln Q(z)\right] \tag{32}$$

$$\leq \ln P(Y), \tag{33}$$

where $F$ and $F'$ are respectively defined as follows:

$$F(z, B_\tau) := \frac{P(Y, B_\tau|\tau(z))}{Q(B_\tau|\tau(z))} P(\tau(z)) R(z|\tau(z)), \quad F'(z, B_\tau) := \frac{F(z, B_\tau)}{Q(z)}. \tag{34}$$

## B.2 Gradient estimators for variational lower bound

The gradient of $\mathcal{L}[Q_{\theta,\phi}, R_\psi]$ with respect to $\theta$ is given by

$$\nabla_\theta \mathcal{L} = \nabla_\theta \mathbb{E}_{Q_\theta(z)} \left[ \mathbb{E}_{Q_\phi(B_\tau|z)}[\ln F(z, B_\tau)] - \ln Q_\theta(z) \right] \tag{35}$$

$$= \mathbb{E}_{Q_\theta(z)} \left[ (\nabla_\theta \ln Q_\theta(z)) \left( \mathbb{E}_{Q_\phi(B_\tau|\tau(z))} \left[ \ln \frac{P(Y, B_\tau|\tau(z))}{Q_\phi(B_\tau|\tau(z))} \right] + \ln P(\tau(z)) R_\psi(z|\tau(z)) \right) \right]$$
$$+ \nabla_\theta \mathbb{H}[Q_\theta(z)], \tag{36}$$

where $\mathbb{H}$ denotes the differential entropy. We assume that $Q_\phi(B_\tau|\tau)$ is reparameterizable as in [40]: namely, $B_\tau$ can be sampled through $B_\tau = h_\phi(\epsilon_B, \tau)$, where $\epsilon_B \sim p_B(\epsilon_B)$, where $p_B(\epsilon)$ and $h_\phi$ denote a parameter-free base distribution and a differentiable function with $\phi$, respectively. Consequently, the gradient of $\mathcal{L}$ with respect to $\phi$ is evaluated as follows:

$$\nabla_\phi \mathcal{L} = \nabla_\phi \mathbb{E}_{Q_\theta(z)} \mathbb{E}_{Q_\phi(B_\tau|z)}[\ln F(z, B_\tau)] \tag{37}$$

$$= \mathbb{E}_{Q_\theta(z)} \mathbb{E}_{p_B(\epsilon_B)} \left[ \nabla_\phi \ln \frac{P(Y, B_\tau = h_\phi(\epsilon_B, \tau)|\tau(z))}{Q_\phi(B_\tau = h_\phi(\epsilon_B, \tau)|\tau(z))} \right]. \tag{38}$$

Lastly, the gradient of $\mathcal{L}$ with respect to $\psi$ can be evaluated with a tractable density model $R_\psi(z|\tau)$ as follows:

$$\nabla_\psi \mathcal{L} = \nabla_\psi \mathbb{E}_{Q_\theta(z)} \mathbb{E}_{Q_\phi(B_\tau|z)}[\ln F(z, B_\tau)] = \mathbb{E}_{Q_\theta(z)} \left[ \nabla_\psi \ln R_\psi(z|\tau(z)) \right]. \tag{39}$$

Given samples $\epsilon_z \sim p_z$ and $\epsilon_B \sim p_B$, we can compute $z = h_\theta(\epsilon_z)$, $\tau(z)$, and $B_\tau = h_\phi(\epsilon_B, \tau(z))$. Then, the below equations are estimators of gradients $\nabla_\theta \mathcal{L}$, $\nabla_\phi \mathcal{L}$, and $\nabla_\psi \mathcal{L}$, respectively:

$$\widehat{g}_\theta = \nabla_\theta \ln Q_\theta(z) \cdot \ln F(z, B_\tau) - \nabla_\theta \ln Q_\theta(h_\theta(\epsilon_z)), \tag{40}$$

$$\widehat{g}_\phi = \nabla_\phi \ln F(z, h_\phi(\epsilon_B, \tau(z))) = \nabla_\phi \ln \frac{P(Y, h_\phi(\epsilon_B, \tau(z))|\tau(z))}{Q_\phi(h_\phi(\epsilon_B, \tau(z))|\tau(z))}, \tag{41}$$

$$\widehat{g}_\psi = \nabla_\psi \ln F(z, h_\phi(\epsilon_B, \tau(z))) = \nabla_\psi \ln R_\psi(z|\tau(z)). \tag{42}$$

The gradients can be computed through the auto-gradient of the following target:

$$\widehat{\mathcal{L}}' = \ln Q_\theta(z) \cdot \text{detach}[f(z, B_\tau)] + f(z, h_\phi(\epsilon_B, \tau(z))) - \ln Q_\theta(h_\theta(\epsilon_z)), \tag{43}$$

where we denote $f(z, B_\tau) = \ln F(z, B_\tau)$, and $\text{detach}[\cdot]$ refers to an operation that blocks backpropagation through its argument. For clarity in terms of differentiability with respect to the parameters, we distinguish between expressions $(z, B_\tau)$ and $(h_\theta(\epsilon_z), h_\phi(\epsilon_B, \tau(z)))$.

## B.3 Multi-sample gradient estimators

Given a $K$ set of Monte Carlo (MC) samples from $Q_{\theta,\phi}(z, B_\tau)$, i.e. $\{\epsilon_Z^{(k)}, z^{(k)} = h_\theta(\epsilon_Z^{(k)})\}_{k=1}^K$ and $\{\epsilon_B^{(k)}, B_\tau^{(k)} = h_\phi(\epsilon_B^{(k)}, \tau(z^{(k)}))\}_{k=1}^K$, we can simply estimate $\nabla L_\theta[Q_{\theta,\phi}, R_\psi]$ as follows:

$$\widehat{g}_\theta^{(K)} = \frac{1}{K} \sum_{k=1}^K \left( \nabla_\theta \ln Q_\theta(z^{(k)}) \cdot f(z^{(k)}, B_\tau^{(k)}) - \nabla_\theta \ln Q_\theta(h_\theta(\epsilon_z^{(k)})) \right). \tag{44}$$

As a simple extension of equation (43), the gradients are obtained through an auto-gradient computation of the following target:

$$\widehat{\mathcal{L}}'^{(K)} = \frac{1}{K} \sum_{k=1}^K \left( \ln Q_\theta(z^{(k)}) \cdot \text{detach}[f(z^{(k)}, B_\tau^{(k)})] + f(z^{(k)}, h_\phi(\epsilon_B^{(k)}, \tau(z^{(k)}))) - \ln Q_\theta(h_\theta(\epsilon_z^{(k)})) \right),$$
$$\tag{45}$$

## B.4 Leave-one-out (LOO) control variates for variance reduction

For the term of $K$-sample gradient estimator $\widehat{g}_\theta^{(K)}$ proportional to the score function $\nabla_\theta \ln Q_\theta$, a leave-one-out (LOO) variance reduction is known to be effective [15, 28], which is denoted as follows:

$$\widehat{g}_{\text{LOO},\theta}^{(K)} = \frac{1}{K} \sum_{k=1}^K \left[ \nabla_\theta \ln Q_\theta(z^{(k)}) \cdot \left( f(z^{(k)}, B_\tau^{(k)}) - \overline{f_k}(z^{(\backslash k)}, B_\tau^{(\backslash k)}) \right) - \nabla_\theta \ln Q_\theta(h_\theta(\epsilon_z^{(k)})) \right],$$
$$\tag{46}$$

where $\overline{f_k}$ denotes:

$$\overline{f_k}(z^{(\backslash k)}, B_\tau^{(\backslash k)}) := \frac{1}{K-1} \sum_{k'=1, k' \neq k}^{K} f(z^{(k')}, B_\tau^{(k')}). \tag{47}$$

To employ the LOO gradient estimator for $\theta$, the target of auto-gradient computation in equation (45) needs to be adjusted as follows:

$$\widehat{\mathcal{L}}'^{(K)}_{\text{LOO}} = \frac{1}{K} \sum_{k=1}^{K} \Big( \ln Q_\theta(z^{(k)}) \cdot \text{detach}[f(z^{(k)}, B_\tau^{(k)}) - \overline{f_k}(z^{(\backslash k)}, B_\tau^{(\backslash k)})]$$

$$+ f(z^{(k)}, h_\phi(\epsilon_B^{(k)}, \tau(z^{(k)}))) - \ln Q_\theta(h_\theta(\epsilon_z^{(k)})) \Big), \tag{48}$$

## B.5 LAX estimators for adaptive variance reduction

---
**Algorithm 2** GeoPhy algorithm with LAX
---
1: $\theta, \phi, \psi, \chi \leftarrow$ Initialize variational parameters
2: **while** not converged **do**
3:      $\epsilon_z^{(1:K)}, \epsilon_B^{(1:K)} \leftarrow$ Random samples from distributions $p_z, p_B$
4:      $z^{(1:K)} \leftarrow h_\theta(\epsilon_z^{(1:K)})$
5:      $B_\tau^{(1:K)} \leftarrow h_\phi(\epsilon_B^{(1:K)}, \tau(z^{(1:K)}))$
6:      $\widehat{g}_\theta, \widehat{g}_\phi, \widehat{g}_\psi \leftarrow$ Estimate the gradients $\nabla_\theta \mathcal{L}, \nabla_\phi \mathcal{L}, \nabla_\psi \mathcal{L}$
7:      $\widehat{g}_\chi \leftarrow \nabla_\chi \left\langle \widehat{g}_\theta^2 \right\rangle$
8:      $\theta, \phi, \psi, \chi \leftarrow$ Update parameters using an SGA algorithm, given gradients $\widehat{g}_\theta, \widehat{g}_\phi, \widehat{g}_\psi, \widehat{g}_\chi$
9: **end while**
10: **return** $\theta, \phi, \psi$
---

The LAX estimator [7] is a stochastic gradient estimator based on a surrogate function, which can be adaptively learned to reduce the variance regarding the term $\nabla_\theta \ln Q_\theta(z)$. In our case, the LAX estimator is given as follows:

$$\widehat{g}_{\text{LAX},\theta} := \nabla_\theta \ln Q_\theta(z) \cdot (f(z, B_\tau) - s_\chi(z)) + \nabla_\theta s_\chi(h_\theta(\epsilon_z)). \tag{49}$$

We summarize the modified procedure of GeoPhy with LAX in Algorithm 2. As we assume $Q_\theta(z)$ is differentiable with respect to $z$, we can also use a modified estimator as follows:

$$\widehat{g}_{\text{LAX},\theta} := \nabla_\theta \ln Q_\theta(z) \cdot (f(z, B_\tau) - s_\chi(z)) + \nabla_\theta s_\chi(h_\theta(\epsilon_z)) - \nabla_\theta \ln Q_\theta(h_\theta(\epsilon_z)). \tag{50}$$

Since it is favorable to reduce the variance of $\widehat{g}_{\text{LAX},\theta}$, we optimize $\chi$ to minimize the following objective as proposed in [7]:

$$\left\langle \mathbb{V}_{Q_\theta(z)}[\widehat{g}_\theta] \right\rangle := \frac{1}{n_\theta} \sum_{i=1}^{n_\theta} \mathbb{V}_{Q_\theta(z)}[\widehat{g}_{\theta_i}] = \frac{1}{n_\theta} \sum_{i=1}^{n_\theta} \left( \mathbb{E}_{Q_\theta(z)}[\widehat{g}_{\theta_i}^2] - \mathbb{E}_{Q_\theta(z)}[\widehat{g}_{\theta_i}]^2 \right), \tag{51}$$

where $n_\theta$ denotes the dimension of $\theta$. As the gradient in equation (49) is given as an unbiased estimator of $\nabla_\theta \mathcal{L}$, which is not dependent on $\chi$, we can use the relation $\nabla_\chi \mathbb{E}_{Q_\theta(z)}[\widehat{g}_{\text{LAX},\theta_i}] = 0$. Therefore, the unbiased estimator of the gradient $\nabla_\chi \left\langle \mathbb{V}_{Q_\theta(z)}[\widehat{g}_{\text{LAX},\theta}] \right\rangle$ is given as follows:

$$\widehat{g}_\chi = \frac{1}{n_\theta} \sum_{i=1}^{n_\theta} \nabla_\chi \widehat{g}_{\text{LAX},\theta_i}^2. \tag{52}$$

As we require the gradient of $\nabla_\theta \mathcal{L}$ with respect to $\chi$ for the optimization, we use different objectives for auto-gradient computation with respect to $\theta$ and the other parameters $\phi$ and $\psi$ as follows:

$$\widehat{\mathcal{L}}'_{\text{LAX},\theta} = \ln Q_\theta(z) \cdot (\text{detach}[f(z, B_\tau)] - s_\chi(z)) + s_\chi(h_\theta(\epsilon_z)) - \ln Q_\theta(h_\theta(\epsilon_z)), \tag{53}$$

$$\widehat{\mathcal{L}}'_{\phi,\psi} = f(z, h_\phi(\epsilon_B, \tau(z))). \tag{54}$$

## B.6 LAX estimators with multiple MC-samples

For the cases with $K$ MC-samples, we use LAX estimators by differentiating the following objectives:

$$\widehat{\mathcal{L}}'^{(K)}_{\text{LAX},\theta} = \frac{1}{K} \sum_{k=1}^{K} \left( \ln Q_\theta(z^{(k)}) \cdot \left( \text{detach}[f(z^{(k)}, B_\tau^{(k)})] - s_\chi(z^{(k)}) \right) + s_\chi(h_\theta(\epsilon_z^{(k)})) - \ln Q_\theta(h_\theta(\epsilon_z^{(k)})) \right),$$ 
(55)

$$\widehat{\mathcal{L}}'^{(K)}_{\phi,\psi} = \frac{1}{K} \sum_{k=1}^{K} f(z^{(k)}, h_\phi(\epsilon_B^{(k)}, \tau(z^{(k)}))).$$ 
(56)

When we combine LAX estimators with LOO control variates. the target for auto-gradient computation changes to the following:

$$\widehat{\mathcal{L}}'^{(K)}_{\text{LOO+LAX},\theta} = \frac{1}{K} \sum_{k=1}^{K} \left( \ln Q_\theta(z^{(k)}) \cdot \left( \text{detach}[f(z^{(k)}, B_\tau^{(k)}) - \overline{f_k}(z^{(\backslash k)}, B_\tau^{(\backslash k)})] - s_\chi(z^{(k)}) \right) \right.$$
$$\left. + s_\chi(h_\theta(\epsilon_z^{(k)})) - \ln Q_\theta(h_\theta(\epsilon_z^{(k)})) \right).$$ 
(57)

We note that $\widehat{\mathcal{L}}'^{(K)}_{\phi,\psi}$ is not affected by the introduction of LOO control variates.

## B.7 Derivation of importance-weighted evidence lower bound (IW-ELBO)

An importance-weighted evidence lower bound (IW-ELBO) [1], is a tighter lower bound of the log-likelihood $\ln P(Y)$ than ELBO. For our model, a conventional $K$-sample IW-ELBO is given as follows:

$$\mathcal{L}^{(K)}_{\text{IW}}[Q] := \mathbb{E}_{Q(z^{(1)}, B_\tau^{(1)}) \cdots Q(z^{(K)}, B_\tau^{(K)})} \left[ \ln \frac{1}{K} \sum_{k=1}^{K} \frac{P(Y, B_\tau^{(k)}, \tau(z^{(k)}))}{Q(B_\tau^{(k)}, \tau(z^{(k)}))} \right].$$ 
(58)

The fact that $\mathcal{L}^{(K)}_{\text{IW}}[Q]$ is the lower bound of $\ln P(Y)$ is directly followed from Theorem 1 in [1]. However, as our model cannot directly evaluate the mass function $Q(\tau)$, we must resort to considering the second lower bound, similar to the case of $K = 1$ as depicted in Proposition 1. We define the $K$-sample tractable IW-ELBO as follows:

$$\mathcal{L}^{(K)}_{\text{IW}}[Q, R] := \mathbb{E}_{Q(z^{(1)}, B_\tau^{(1)}) \cdots Q(z^{(K)}, B_\tau^{(K)})} \left[ \ln \frac{1}{K} \sum_{k=1}^{K} F'(z^{(k)}, B_\tau^{(k)}) \right],$$ 
(59)

where $F'$ is defined in equation (34). We will prove in Theorem 1 that $\mathcal{L}^{(K)}_{\text{IW}}[Q, R]$ serves as a lower bound of the $\ln P(Y)$. Although this inequality holds when $K = 1$, as shown by $\ln P(Y) \geq \mathcal{L}[Q] \geq \mathcal{L}[Q, R]$ in Proposition 1, the relationship is less obvious when $K > 1$. Before delving into that, we prepare the following proposition.

**Proposition 2.** Given $Q(z, \tau)$ as defined in equation 4 and an arbitrary conditional distribution $R(z|\tau)$, it follows that

$$\mathbb{E}_{R(z|\tau)}[\mathbb{I}[\tau = \tau(z)]] \leq 1,$$ 
(60)

where setting $R(z|\tau) = Q(z|\tau)$ is a sufficient condition for the equality to hold.

*Proof.* The inequality immediately follows from the definition as follows:

$$\mathbb{E}_{R(z|\tau)}[\mathbb{I}[\tau = \tau(z)]] \leq \mathbb{E}_{R(z|\tau)}[1] = 1.$$ 
(61)

Next, when we set $R(z|\tau) = Q(z|\tau)$, the condition for equality is satisfied as follows:

$$\mathbb{E}_{Q(z|\tau)}[\mathbb{I}[\tau = \tau(z)]] = \frac{\mathbb{E}_{Q(z)}[\mathbb{I}[\tau = \tau(z)]^2]}{Q(\tau)} = \frac{Q(\tau)}{Q(\tau)} = 1,$$ 
(62)

where we have used the definition of $Q(\tau) := \mathbb{E}_{Q(z)}[\mathbb{I}[\tau = \tau(z)]]$ from equation (4) and the resulting relation $Q(z|\tau)Q(\tau) = \mathbb{I}[\tau = \tau(z)]Q(z)$.  □

**Theorem 1.** Given $Q(z, \tau)$ as defined in equation 4 and an arbitrary conditional distribution $R(z|\tau)$ that satisfies $\mathrm{supp}R(z|\tau) \supseteq \mathrm{supp}Q(z|\tau)$, for any natural number $K > 1$, the following relation holds:

$$\ln P(Y) \geq \mathcal{L}_{\mathrm{IW}}^{(K)}[Q, R] \geq \mathcal{L}_{\mathrm{IW}}^{(K-1)}[Q, R]. \tag{63}$$

Additionally, if $F'(z, B_\tau)$ is bounded and $\forall \tau, \mathbb{E}_{R(z|\tau)}[\mathbb{I}[\tau = \tau(z)]] = 1$, then $\mathcal{L}_{\mathrm{IW}}^{(K)}[Q, R]$ approaches $\ln P(Y)$ as $K \to \infty$.

*Proof.* We first show that for any natural number $K > M$,

$$\mathcal{L}_{\mathrm{IW}}^{(K)}[Q, R] \geq \mathcal{L}_{\mathrm{IW}}^{(M)}[Q, R]. \tag{64}$$

For simplicity, we denote $Q(z^{(k)}, B_\tau^{(k)})$ and $F'(z^{(k)}, B_\tau^{(k)})$ as $Q_k$ and $F'_k$, respectively, in the following discussion. Let $U_M^K$ represent a uniform distribution over a subset with $M$ distinct indices chosen from the $K$ indices $\{1, \ldots, K\}$. Similar to the approach used in [1], we will utilize the following relationship:

$$\frac{1}{K} \sum_{k=1}^{K} F'_k = \mathbb{E}_{\{i_1, \ldots, i_M\} \sim U_M^K} \left[ \frac{1}{M} \sum_{m=1}^{M} F'_{i_m} \right]. \tag{65}$$

Now, the inequality (64) is derived as follows:

$$\mathbb{E}_{Q_1 \cdots Q_K} \left[ \ln \left( \frac{1}{K} \sum_{k=1}^{K} F'_k \right) \right] = \mathbb{E}_{Q_1 \cdots Q_K} \left[ \ln \mathbb{E}_{\{i_1, \ldots, i_m\} \sim U_M^K} \left[ \left( \frac{1}{M} \sum_{m=1}^{M} F'_{i_m} \right) \right] \right] \tag{66}$$

$$\geq \mathbb{E}_{Q_1 \cdots Q_K} \left[ \mathbb{E}_{\{i_1, \ldots, i_m\} \sim U_M^K} \left[ \ln \left( \frac{1}{M} \sum_{m=1}^{M} F'_{i_m} \right) \right] \right] \tag{67}$$

$$= \mathbb{E}_{Q_1 \cdots Q_M} \left[ \ln \left( \frac{1}{M} \sum_{m=1}^{M} F'_m \right) \right], \tag{68}$$

where we have also used Jensen's inequality.

Next, we show that $\ln P(Y) \geq \mathcal{L}_{\mathrm{IW}}^{(K)}[Q, R]$. We again use Jensen's inequality as follows:

$$\mathcal{L}_{\mathrm{IW}}^{(K)}[Q, R] = \mathbb{E}_{Q_1 \cdots Q_K} \left[ \ln \frac{1}{K} \sum_{k=1}^{K} F'_k \right] \tag{69}$$

$$\leq \ln \mathbb{E}_{Q_1 \cdots Q_K} \left[ \frac{1}{K} \sum_{k=1}^{K} F'_k \right] = \ln \mathbb{E}_{Q(z, B_\tau)} \left[ F'(z, B_\tau) \right]. \tag{70}$$

The last term is further transformed as follows:

$$\ln \mathbb{E}_{Q(z, B_\tau)} \left[ F'(z, B_\tau) \right] = \ln \mathbb{E}_{Q(z, B_\tau)} \left[ \frac{P(Y, B_\tau | \tau(z)) R(z | \tau(z))}{Q(B_\tau | \tau(z)) Q(z)} \right] \tag{71}$$

$$= \ln \mathbb{E}_{Q(z, B_\tau)} \left[ \frac{P(Y, B_\tau | \tau(z)) R(z | \tau(z))}{Q(z, B_\tau)} \right] \tag{72}$$

$$= \ln \mathbb{E}_{Q(z)} \left[ \frac{P(Y, \tau(z)) R(z | \tau(z))}{Q(z)} \right] \tag{73}$$

$$= \ln \sum_{\tau' \in \mathcal{T}} \mathbb{E}_{Q(z)} \left[ \frac{P(Y, \tau') R(z | \tau')}{Q(z)} \mathbb{I}[\tau' = \tau(z)] \right] \tag{74}$$

$$= \ln \sum_{\tau' \in \mathcal{T}} P(Y, \tau') \mathbb{E}_{R(z | \tau')} \left[ \mathbb{I}[\tau' = \tau(z)] \right] \tag{75}$$

$$\leq \ln \sum_{\tau' \in \mathcal{T}} P(Y, \tau') = \ln P(Y), \tag{76}$$

where, in the transition from the first to the second row, we employed the following relation:

$$Q(B_\tau|\tau(z)) = \sum_{\tau \in \mathcal{T}} Q(B_\tau|\tau)\mathbb{I}[\tau = \tau(z)] = \sum_{\tau \in \mathcal{T}} Q(B_\tau|\tau)Q(\tau|z) = Q(B_\tau|z), \qquad (77)$$

and we have used Proposition 2 for the last inequality.

Finally, we will show that the following convergence property assuming that $F(z, B_\tau)$ is bounded:

$$\mathcal{L}_{\text{IW}}^{(K)}[Q, R] \to \ln\left(\sum_{\tau' \in \mathcal{T}} P(Y, \tau')\, \mathbb{E}_{R(z|\tau')}\, \mathbb{I}[\tau' = \tau(z)]\right) \qquad (K \to \infty). \qquad (78)$$

From the strong law of large numbers, it follows that $\frac{1}{K}\sum_{k=1}^{K} F'_k$ converges to the following term almost surely as $K \to \infty$:

$$\mathbb{E}_{Q(z_k, B_k)}\left[F'(z_k, B_k)\right] = \sum_{\tau' \in \mathcal{T}} P(Y, \tau')\, \mathbb{E}_{R(z|\tau')}\, \mathbb{I}[\tau' = \tau(z)], \qquad (79)$$

where we have employed the same transformations as used from equation (71) to (75). Observe that the *r.h.s* term of the equation (78) equals to $\ln P(Y)$ when $\forall \tau' \in \mathcal{T}, \mathbb{E}_{R(z|\tau')}[\mathbb{I}[\tau' = \tau(z)]] = 1$, which completes the proof. $\qquad\square$

**Estimation of marginal log-likelihood** For the estimation of $\ln P(Y)$, we employ $\mathcal{L}^{(K)}[Q, R]$ with $K = 1,000$ similar to [40]. From Theorem 1, IW-ELBO $\mathcal{L}^{(K)}[Q, R]$ is at least a better lower bound of $\ln P(Y)$ than ELBO $\mathcal{L}[Q, R]$, and converges to $\ln P(Y)$ when $\forall \tau, \mathbb{E}_{R(z|\tau)}[\mathbb{I}[\tau = \tau(z)]] = 1$. According to Proposition 2, this equality condition is satisfied when we set $R(z|\tau) = Q(z|\tau)$, which is approached by maximizing $\mathcal{L}[Q, R]$ with respect to $R$ as indicated in Proposition 1.

## B.8 Gradient estimators for IW-ELBO

The gradient of IW-ELBO $\mathcal{L}_{\text{IW}}^{(K)}[Q_{\theta,\phi}, R_\psi]$ with respect to $\theta$ is given by

$$\nabla_\theta \mathcal{L}_{\text{IW}}^{(K)} = \mathbb{E}_{Q_{\theta,\phi}(z^{(1)}, B_\tau^{(1)})\cdots Q_{\theta,\phi}(z^{(K)}, B_\tau^{(K)})}\left[\sum_{k=1}^{K} w_k(z^{(1:K)}, B^{(1:K)})\nabla_\theta \ln F'(z^{(k)}, B_\tau^{(k)})\right]$$

$$+ \mathbb{E}_{Q_{\theta,\phi}(z^{(1)}, B_\tau^{(1)})\cdots Q_{\theta,\phi}(z^{(K)}, B_\tau^{(K)})}\left[\sum_{k=1}^{K} \nabla_\theta \ln Q_\theta(z^{(k)}) \cdot \ell(z^{(1:K)}, B^{(1:K)})\right], \qquad (80)$$

where we have defined

$$w_k(z^{(1:K)}, B_\tau^{(1:K)}) := \frac{F'(z^{(k)}, B_\tau^{(k)})}{\sum_{k'=1}^{K} F'(z^{(k')}, B_\tau^{(k')})}, \qquad (81)$$

$$\ell(z^{(1:K)}, B_\tau^{(1:K)}) := \ln\left(\frac{1}{K}\sum_{k'=1}^{K} F'(z^{(k')}, B_\tau^{(k')})\right). \qquad (82)$$

Similarly, as $F'$ is differentiable with respect to $B_\tau$, and $B_\tau = h_\phi(\epsilon_B, \tau)$ is differentiable with respect to $\phi$, the gradient of $\mathcal{L}_{\text{IW}}^{(K)}$ with respect to $\phi$ can be evaluated as follows:

$$\nabla_\phi \mathcal{L}_{\text{IW}}^{(K)} = \mathbb{E}_{Q_\theta(z^{(1)})\cdots Q_\theta(z^{(K)})}\, \mathbb{E}_{p_B(\epsilon_B^{(1)})\cdots p_B(\epsilon_B^{(K)})}\left[\sum_{k=1}^{K} w_k(z^{(1:K)}, B_\tau^{(1:K)})\nabla_\phi \ln F'(z^{(k)}, h_\phi(\epsilon^{(k)}, \tau))\right].$$
$$\qquad (83)$$

Since $\nabla_\theta \ln F'(z^{(k)}, B_\tau^{(k)}) = -\nabla_\theta \ln Q_\theta(z^{(k)})$ from equation (34), an unbiased estimator of the gradient $\nabla_\theta \mathcal{L}^{(K)}$ is given as follows:

$$\widehat{g}_{\text{IW},\theta}^{(K)} := \sum_{k=1}^{K} \nabla_\theta \ln Q_\theta(z^{(k)}) \cdot \left[-w_k(z^{(1:K)}, B_\tau^{(1:K)}) + \ell(z^{(1:K)}, B_\tau^{(1:K)})\right]. \qquad (84)$$

The remaining gradient estimators are given as follows:

$$\widehat{g}_{\mathrm{IW},\phi}^{(K)} = \sum_{k=1}^{K} w_k(z^{(1:K)}, B_\tau^{(1:K)}) \nabla_\phi \ln F(z^{(k)}, h_\phi(\epsilon^{(k)}, \tau)), \tag{85}$$

$$\widehat{g}_{\mathrm{IW},\phi}^{(K)} = \sum_{k=1}^{K} w_k(z^{(1:K)}, B_\tau^{(1:K)}) \nabla_\psi \ln F(z^{(k)}, h_\phi(\epsilon^{(k)}, \tau)). \tag{86}$$

In total, the target for computing auto-gradient for these gradients is given as follows:

$$\widehat{\mathcal{L}}_{\mathrm{IW}}^{\prime(K)} = \sum_{k=1}^{K} \ln Q_\theta(z^{(k)}) \cdot \mathrm{detach}\left[-w_k(z^{(1:K)}, B_\tau^{(1:K)}) + \ell(z^{(1:K)}, B_\tau^{(1:K)})\right]$$

$$+ \sum_{k=1}^{K} \mathrm{detach}[w_k(z^{(1:K)}, B_\tau^{(1:K)})] \ln F(z^{(k)}, h_\phi(\epsilon_B^{(k)}, \tau(z^{(k)}))). \tag{87}$$

### B.9  VIMCO estimators

The VIMCO estimator [22] aims at reducing the variance in each of the terms proportional to $\ell(z^{(1:K)}, B^{(1:K)})$ in equation (84). This is accomplished by forming for a control variate $\ell_k(z^{(\backslash k)}, B^{(\backslash B_\tau)})$ that consists of only variables other than $z^{(k)}$ and $B^{(k)}$ for every $k$ as follows:

$$\widehat{g}_{\theta,\mathrm{VIMCO}}^{(K)} := \sum_{k=1}^{K} \nabla_\theta \ln Q_\theta(z^{(k)}) \cdot \left[-w_k(z^{(1:K)}, B_\tau^{(1:K)}) + \ell(z^{(1:K)}, B_\tau^{(1:K)}) - \overline{\ell}_k(z^{(\backslash k)}, B_\tau^{(\backslash k)})\right], \tag{88}$$

where we denote that

$$\overline{\ell}_k(z^{(\backslash k)}, B^{(\backslash k)}) := \ln\left(\frac{1}{K}\left(\overline{F'}_k(z^{(\backslash k)}, B_\tau^{(\backslash k)}) + \sum_{k'=1, k'\neq k}^{K} F'(z^{(k')}, B_\tau^{(k')})\right)\right), \tag{89}$$

$$\overline{F'}_k(z^{(\backslash k)}, B_\tau^{(\backslash k)}) := \exp\left(\frac{1}{K-1} \sum_{k'=1, k'\neq k}^{K} \ln F'(z^{(k')}, B_\tau^{(k')})\right). \tag{90}$$

Note that the unbiasedness of $\widehat{g}_{\theta,\mathrm{VIMCO}}$ can be confirmed through the following observation:

$$\mathbb{E}_{Q_{\theta,\phi}(z^{(k)}, B_\tau^{(k)})}\left[\nabla_\theta \ln Q_\theta(z^{(k)}) \cdot \overline{\ell}_k(z^{(\backslash k)}, B_\tau^{(\backslash k)})\right]$$

$$= \overline{\ell}_k(z^{(\backslash k)}, B_\tau^{(\backslash k)}) \cdot \mathbb{E}_{Q_{\theta,\phi}(z^{(k)}, B_\tau^{(k)})}\left[\nabla_\theta \ln Q_\theta(z^{(k)})\right]$$

$$= \overline{\ell}_k(z^{(\backslash k)}, B_\tau^{(\backslash k)}) \cdot \mathbb{E}_{Q_{\theta,\phi}(z^{(k)})}\left[\nabla_\theta \ln Q_\theta(z^{(k)})\right] = 0. \tag{91}$$

To employ the VIMCO estimator for $\theta$, the target for auto-gradient computation as stated in equation (87) needs to be adjusted as follows:

$$\widehat{\mathcal{L}}_{\mathrm{VIMCO}}^{\prime(K)} = \sum_{k=1}^{K} \ln Q_\theta(z^{(k)}) \cdot \mathrm{detach}\left[-w_k(z^{(1:K)}, B_\tau^{(1:K)}) + \ell(z^{(1:K)}, B_\tau^{(1:K)}) - \overline{\ell}_k(z^{(\backslash k)}, B_\tau^{(\backslash k)})\right]$$

$$+ \sum_{k=1}^{K} \mathrm{detach}[w_k(z^{(1:K)}, B_\tau^{(1:K)})] \ln F(z^{(k)}, h_\phi(\epsilon_B^{(k)}), \tau(z^{(k)})). \tag{92}$$

## C  Experimental Details

### C.1  Training process

For the training of GeoPhy, we continued the stochastic gradient descent process until a total of 1,000,000 Monte Carlo (MC) tree samples were consumed. Specifically, if $K$ MC-samples were used

per step, we performed up to 1,000,000 / K steps. It is noteworthy that the number of MC samples equaled the number of likelihood evaluations (NLEs), which provided us with a basis for comparing convergence speed between different runs, as shown in Fig. 2. In all of our experiments, we used Adam optimizer with an initial learning rate of $0.0001$. The learning rate was then multiplied by $0.75$ after every 200,000 steps. Similar to approaches taken by Zhang and Matsen IV [42], we incorporated an annealing procedure during the initial consumption of 100,000 MC samples. Specifically, we replaced the likelihood function in the lower bound with $P(Y|B_\tau, \tau)^\beta$ and linearly increased the inverse temperature $\beta$ from 0.001 to 1 throughout the iterations. Note that all the estimations of marginal log-likelihood (MLL) were performed with $\beta$ set to 1.

For each estimation of MLL values, GeoPhy and VBPI-GNN used IW-ELBO with 1000 samples, whereas all CSMC-based methods used 2048 particle weights [16].

## C.2 Variational branch length distribuions

For the variational branch length distribution $Q_\phi(B_\tau|\tau)$, we followed an architecture of [40]; namely, each branch length independently followed a lognormal distribution which was parameterized with a graph neural network (GNN). Details are described in Appendix A.2.

## C.3 LAX estimators

As input features of a surrogate function $s_\chi(z)$ used in the LAX estimators, we employed a flattened vector of coordinates $z \in \mathbb{R}^{N \times d}$ when $z$ resides in Euclidean space. In cases where the coordinates were $z \in \mathbb{H}^d$, we first transformed $z$ with a logarithm map $\log_{\mu^\circ} z \in T_{\mu^\circ}\mathbb{H}^d$, then omitted their constant value 0-th elements and subsequently flattened the result. We implemented a simple multi-layer perceptron (MLP) network with a single hidden layer of width $10Nd$ and a subsequent sigmoid linear unit (SiLU) activation function as the neural network to output $s_\chi(z)$.

## C.4 Replication of MLL estimates with MrBayes SS

Given the observed discrepancies in marginal log-likelihood (MLL) estimates obtained with the MrBayes stepping-stone (SS) method between references [42] and [16], we replicated the MrBayes SS runs using MrBayes version 3.2.7a. The script we used is provided below.

```
BEGIN MRBAYES;
set autoclose=yes nowarn=yes Seed=123 Swapseed=123;
lset nst=1;
prset statefreqpr=fixed(equal);
prset brlenspr=Unconstrained:exp(10.0);
ss ngen=10000000 nruns=10 nchains=4 printfreq=1000 samplefreq=100 \
savebrlens=yes filename=mrbayes_ss_out;
END;
```

We incorporated the results in the row named Replication in Table 3, where the values aligned more closely with those found in [42]. We deduced that the prior distribution used in [16] might have been set differently as the current default values of brlenspr are Unconstrained : GammaDir$(1.0, 0.100, 1.0, 1.0)$ [3], which deviates from the model assumption used for the benchmarks. We observed that the line brlenspr was not included in the code provided in Appendix F of [16]. Having been able to replicate the results found in [42], we opted to use their values as a reference in Table 1.

## C.5 Visualization of tree topologies

In Fig. 3, we visualized the sum of the probability densities for tip node distribution $\sum_{i=1}^{N} Q(z_i)$ by projecting each hyperbolic coordinate $z_i \in \mathbb{H}^d$ onto the Poincaré coordinates $\overline{z}_{ik} = z_{ik}/(1 + z_{i0})$ $(k = 1, \ldots, d)$, then applying a transformation $\overline{z}_i \mapsto \overline{z}_i' = \tanh(a\|\overline{z}_i\|_2) \cdot \overline{z}_i/\|\overline{z}_i\|_2$

---

[3]`https://github.com/NBISweden/MrBayes/blob/develop/doc/manual/Manual_MrBayes_v3.2.pdf`

with $a = 2.1$ to emphasize the central region. To display the density $Q$ in the new coordinates, the Jacobian term was also considered to evaluate the density $Q(\overline{z}_i')$.

For the comparison of consensus tree topologies, we plotted the edges of the tree by connecting each of their end node coordinate pairs with a geodesic line. The coordinate in $\mathbb{H}^d$ of the $i$-th tip node was determined as the location parameter $\mu_i \in \mathbb{H}^d$ of the wrapped normal distribution $Q(z_i) = \mathcal{WN}(z_i; \mu_i, \Sigma_i)$. Let $\xi_u \in \mathbb{H}^d$ denotes the coordinate of an interior node $u$, we defined $\xi_u$ by using the Lorentzian centroid operation $\mathcal{C}$ [18] as follows:

$$\xi_u := \mathcal{C}(\{c_s\}_{s \in \mathcal{S}_\tau(u)}, \{\nu_s\}_{s \in \mathcal{S}_\tau(u)}) = \frac{\widetilde{\xi}_u}{\sqrt{-\left\langle \widetilde{\xi}_u, \widetilde{\xi}_u \right\rangle_L}}, \tag{93}$$

where $\widetilde{\xi}_u := \sum_{s \in \mathcal{S}_\tau(u)} \nu_s c_s$ denote an unnormalized sum of weighted coordinates, $s \in \mathcal{S}_\tau(u)$ denote a subset of tip node indices partitioned by the interior node $u$ in the tree topology $\tau$, $c_s := \mathcal{C}(\{\mu_i\}_{i \in s}, \{1\}_{i \in s})$ denote the Lorentzian centroid of the tip nodes contained in the subset $s$, and $\nu_s = N - |s|$ denote the number of the tip nodes in the complement set of $s$ where $|s|$ represents the number of tip nodes in the subset $s$. As an unrooted tree topology $\tau$ can be identified by the set of tip node partitions introduced by the interior nodes of $\tau$, the same unrooted tree topologies give the same set of interior coordinates $\{\xi_u\}_{u \in V}$ according to equation (93).

# D   Additional Results

## D.1   Marginal log-likelihood estimates throughout iterations for DS1 dataset

In Fig. 2, we highlight several key findings of training processes. Specifically, (1) control variates were crucial for optimizations; (2) The LAX estimator for individual Monte Carlo (MC) samples ($K = 1$) and leave-one-out (LOO) CVs for multiple MC samples were effective; (3) IW-ELBO and VIMCO were not particularly effective when a full diagonal matrix is used for $Q(z)$. In Fig. 5 and 6, we show that these findings hold consistently across various control variates and configurations of $Q(z)$, including, different types of distributions (normal $\mathcal{N}$ and wrapped normal $\mathcal{WN}$), embedding dimensions ($d = 2, 3$, and $4$), and varying numbers of MC samples ($K = 1, 2$, and $3$).

## D.2   Marginal log-likelihood estimates for eight datasets

We present more comprehensive results in Table 3, extending upon the data from Table 1. This table showcases the marginal log-likelihood (MLL) estimates obtained with various GeoPhy configurations and other conventional methods for the datasets DS1-DS8. Once again, GeoPhy demonstrates its superior performance, consistently outperforming CSMC-based approaches that do not require preselection of tree topologies across the majority of configurations and datasets. This reaffirms the stability and excellence of our approach. Additionally, we found that a $Q(z)$ configuration using a 4-dimensional wrapped normal distribution with a full covariance matrix was the most effective among the tested configurations.

## D.3   Analysis of tree topology distributions

To assess the accuracy and expressivity of GeoPhy's tree topology distribution, we contrasted $Q(\tau)$ with MCMC tree samples. For our analysis, we considered experimental runs using LAX estimators with either $K = 1$ or $3$. For a configuration of $Q(z)$, We employed a diagonal or multivariate wrapped normal distribution with 2, 4, 5, and 6 dimensions. We sampled 1000 MC trees from $Q(\tau)$ for each experiment and then deduced the majority consensus tree. As depicted in Fig. 8 across the eight datasets, the Robinson-Foulds (RF) distances between the majority consensus tree of $Q(\tau)$ and the one obtained with MrBayes were well aligned with the MLL values.

To highlight the expressivity limitation of $Q(\tau)$ obtained from a simple independent distribution of $Q(z)$, we compiled three statistics related to the tree topology samples for DS1, DS4, and DS7 datasets (Table 4), wherein we employed the model with the least RF distance for each dataset. For DS1 and DS4, GeoPhy's tree topologies were less diverse than those of MrBayes and VBPI-GNN and had more concentration on the most frequent topology. For more diffused tree DS7, GeoPhy also showed diverse tree samples with a close diversity index to the other two.

Table 2: Comparative results of the mean MLL estimates for DS1 dataset obtained with GeoPhy for different combinations of the distribution $Q(z)$, per-step Monte Carlo samples ($K$), and control variates (CVs). Each of the mean MLL values is obtained from five independent runs with 1000 MC samples at the last steps. Standard deviations are shown in parentheses. The bold and underlined numbers represent the best (highest) values in row-wise and column-wise comparisons, respectively.

| CV | $\mathcal{N}$ (diag) | | | $\mathcal{N}$ (full) | | |
|---|---|---|---|---|---|---|
| | $d=2$ | $d=3$ | $d=4$ | $d=2$ | $d=3$ | $d=4$ |
| K=1 (LAX) | −7126.79 | −7130.11 | −7123.95 | −7129.70 | **−7119.51** | −7120.03 |
| | (10.06) | (10.63) | (12.03) | (6.14) | (10.90) | (11.92) |
| K=2 (LOO) | −7122.84 | −7125.72 | −7121.79 | −7121.23 | −7130.43 | **−7114.92** |
| | (11.96) | (13.08) | (12.52) | (14.62) | (10.82) | (8.32) |
| K=2 (LOO+LAX) | −7129.26 | −7128.18 | −7116.96 | −7121.74 | −7129.45 | **−7115.14** |
| | (8.35) | (9.83) | (10.39) | (10.73) | (10.23) | (8.26) |
| K=3 (LOO) | −7130.60 | −7124.61 | **−7120.88** | −7132.35 | −7128.26 | −7124.62 |
| | (10.66) | (12.21) | (13.15) | (6.89) | (9.80) | (12.33) |
| K=3 (LOO+LAX) | −7128.36 | −7125.66 | **−7119.81** | −7122.76 | −7123.67 | −7123.37 |
| | (9.77) | (10.95) | (11.71) | (10.81) | (11.23) | (11.28) |

| CV | $\mathcal{WN}$ (diag) | | | $\mathcal{WN}$ (full) | | |
|---|---|---|---|---|---|---|
| | $d=2$ | $d=3$ | $d=4$ | $d=2$ | $d=3$ | $d=4$ |
| K=1 (LAX) | −7126.89 | −7133.78 | −7122.10 | −7124.63 | −7119.59 | **−7111.55** |
| | (10.06) | (13.57) | (12.29) | (8.09) | (10.84) | (0.07) |
| K=2 (LOO) | −7121.13 | −7125.60 | −7121.80 | **−7114.99** | −7130.40 | −7116.14 |
| | (13.06) | (13.04) | (12.52) | (4.66) | (10.80) | (10.66) |
| K=2 (LOO+LAX) | −7120.90 | −7120.96 | −7126.49 | −7119.81 | −7129.58 | **−7116.21** |
| | (10.10) | (13.08) | (12.01) | (9.78) | (10.24) | (10.75) |
| K=3 (LOO) | −7130.67 | −7124.54 | −7120.94 | −7125.94 | −7122.85 | **−7119.77** |
| | (10.67) | (12.20) | (13.11) | (13.07) | (11.96) | (11.80) |
| K=3 (LOO+LAX) | −7128.40 | −7125.69 | −7125.78 | **−7115.19** | −7120.96 | −7116.09 |
| | (9.78) | (13.02) | (13.10) | (8.16) | (13.12) | (10.67) |

We also summarize the frequency of bipartitions observed for tree topology samples in Fig. 8. The intermediate values between 0 and 1 in this frequency plot reveal topology diversities, where VBPI-GNN aligns more closely with MrBayes compared to GeoPhy. For DS1 and DS4, GeoPhy tends to take values near zero and one in the slope region. For DS7, while GeoPhy traced the curve more accurately, fluctuations around this curve highlight potential room for improvement.

## D.4 Runtime comparison

We measured CPU runtimes using a single thread on a 20-core Xeon processor. In Fig. 10, we compared single-core runtime of MrBayes, VPBI-GNN, and GeoPhy (ours). As MrBayes is written in C and highly optimized its per-iteration computation time is much faster than current variational methods. In general, the per-step efficiency of GeoPhy is comparable to that of VBPI-GNN. Although the convergence of MLL estimates tends to be slow in terms of the number of consumed samples for larger $K$ (Fig. 9 Left), the total CPU-time slightly decreases as $K$ increases (Fig. 10 Left). We also included the estimated runtime across the eight datasets and the different model configurations, where the number of species ranges from 27 to 64 (Fig. 10 Right).

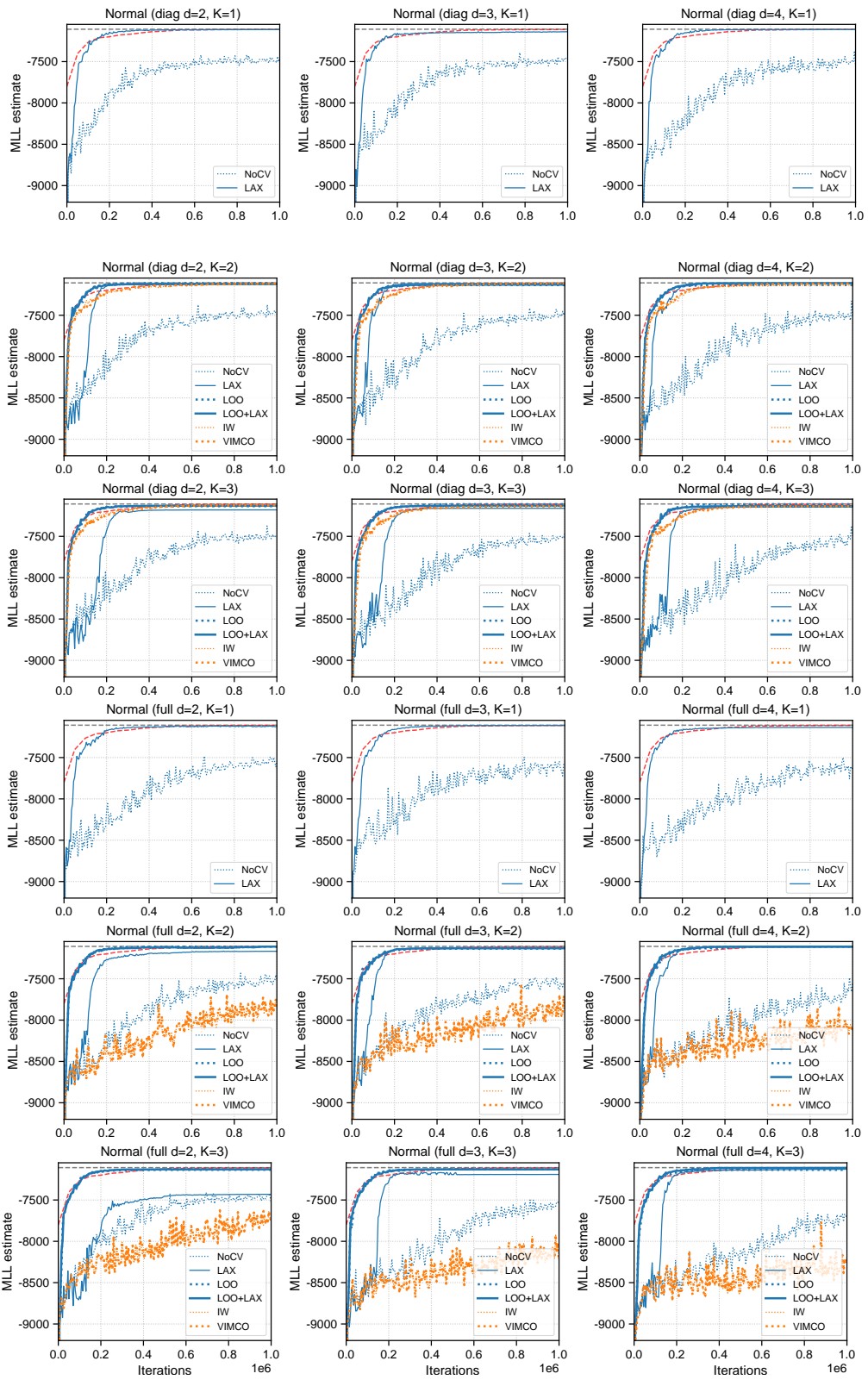

Figure 5: Comparison of marginal log-likelihood (MLL) estimates for DS1 dataset using different control variates and configurations of $Q(z)$. We configured $Q(z)$ to be an independent normal distribution of dimensions $d = 2, 3,$ or $4$ with either a diagonal (diag) or full covariance matrix. The number of MC samples was set to $K = 1, 2,$ or $3$. For a detailed explanation of the legend, refer to the caption of Fig. 2.

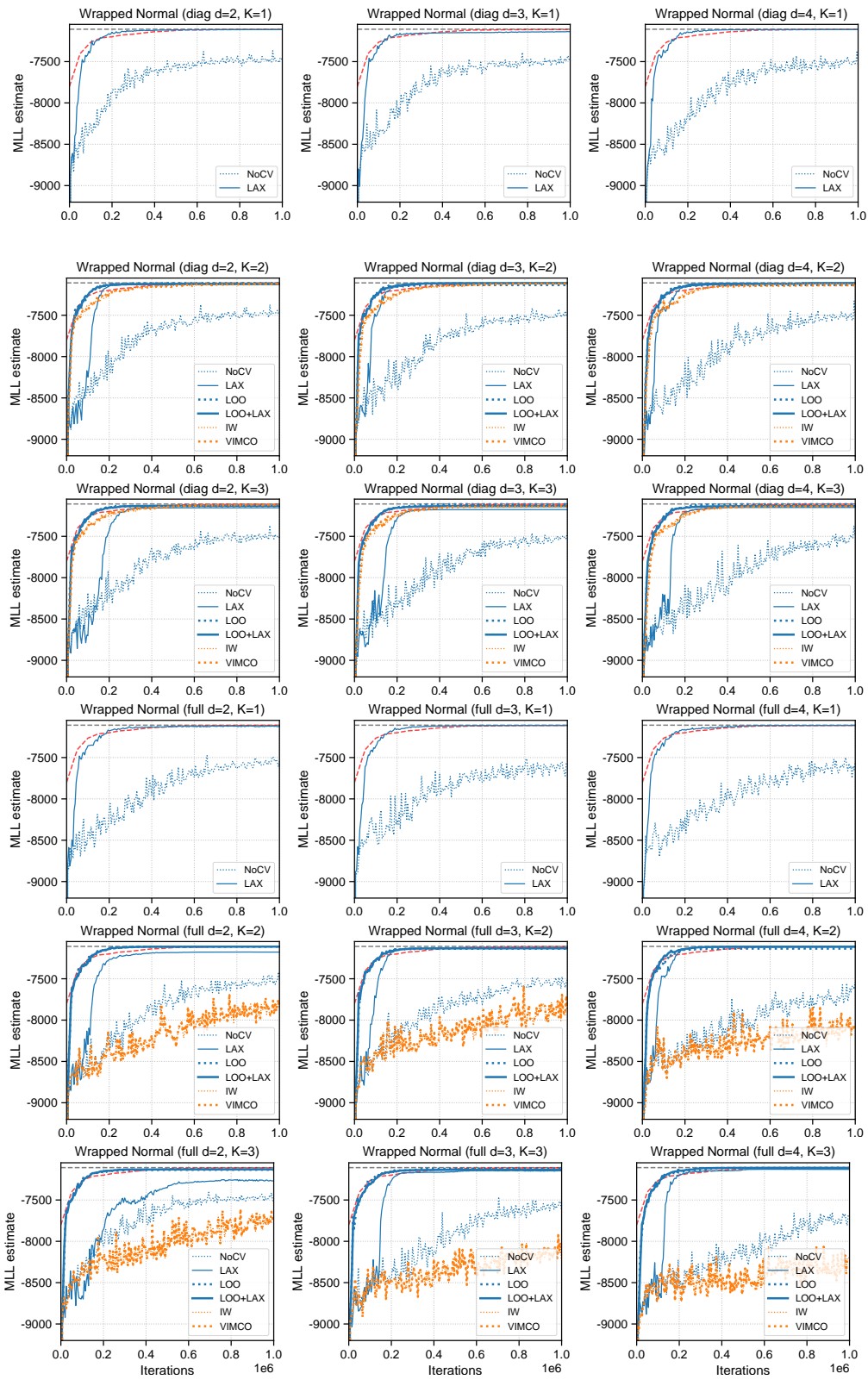

Figure 6: Comparison of marginal log-likelihood (MLL) estimates for DS1 dataset using different control variates and configurations of $Q(z)$. We employed wrapped normal distributions for $Q(z)$ instead of normal distributions shown in Fig. 5. The dimensions $d = 2, 3$, or $4$ and the number of MC samples $K = 1, 2$, or $3$ are the same as in the referenced figure.

Table 3: Extended results of Table 1 comparing the marginal log-likelihood (MLL) estimates with different approaches in eight benchmark datasets. The MLL values for MrBayes SS and VBPI-GNN were obtained from [40], while CSMC, VCSMC, and $\phi$-CSMSC are referenced from [16]. We also included replicated results for MrBayes SS. The MLL values for our approach (GeoPhy) are presented for a variety of $Q(z)$ configurations, encompassing distribution types (normal $\mathcal{N}$ or wrapped normal $\mathcal{WN}$), embedding dimensions (2 or 4), and the covariance matrix (full or diagonal). Each result features various CVs: LAX with $K = 1$, LOO with $K = 3$ denoted as LOO(3), and a combination of LOO and LAX, denoted as LOO(3)+. The figures highlighted in bold represent the highest values obtained with GeoPhy and the three CSMC-based methods, all of which perform an approximate Bayesian inference without the preselection of topologies. We have underlined MLL estimates where GeoPhy outperformed the other CSMC-based methods.

| Dataset | DS1 | DS2 | DS3 | DS4 | DS5 | DS6 | DS7 | DS8 |
|---|---|---|---|---|---|---|---|---|
| Reference | [8] | [6] | [37] | [9] | [17] | [43] | [38] | [30] |
| #Taxa ($N$) | 27 | 29 | 36 | 41 | 50 | 50 | 59 | 64 |
| #Sites ($M$) | 1949 | 2520 | 1812 | 1137 | 378 | 1133 | 1824 | 1008 |
| MrBayes SS | −7108.42 | −26367.57 | −33735.44 | −13330.06 | −8214.51 | −6724.07 | −37332.76 | −8649.88 |
| [29, 42] | (0.18) | (0.48) | (0.5) | (0.54) | (0.28) | (0.86) | (2.42) | (1.75) |
| (Replication) | −7107.81 | −26366.45 | −33732.79 | −13328.40 | −8209.17 | −6721.54 | −37331.85 | −8646.18 |
| | (0.25) | (0.40) | (0.63) | (0.48) | (0.46) | (0.77) | (3.08) | (1.19) |
| VBPI-GNN | −7108.41 | −26367.73 | −33735.12 | −13329.94 | −8214.64 | −6724.37 | −37332.04 | −8650.65 |
| [40] | (0.14) | (0.07) | (0.09) | (0.19) | (0.38) | (0.4) | (0.26) | (0.45) |
| CSMC | −8306.76 | −27884.37 | −35381.01 | −15019.21 | −8940.62 | −8029.51 | − | −11013.57 |
| [34, 16] | (166.27) | (226.6) | (218.18) | (100.61) | (46.44) | (83.67) | − | (113.49) |
| VCSMC | −9180.34 | −28700.7 | −37211.2 | −17106.1 | −9449.65 | −9296.66 | − | − |
| [24, 16] | (170.27) | (4892.67) | (397.97) | (362.74) | (2578.58) | (2046.7) | − | − |
| $\phi$-CSMC | −7290.36 | −30568.49 | −33798.06 | −13582.24 | −8367.51 | −7013.83 | − | −9209.18 |
| [16] | (7.23) | (31.34) | (6.62) | (35.08) | (8.87) | (16.99) | − | (18.03) |
| $\mathcal{N}$(diag,2) | −7126.79 | −26440.54 | −33814.98 | −13356.21 | −8251.99 | −6747.15 | −37526.41 | −8727.93 |
| | (10.06) | (28.78) | (20.31) | (9.48) | (9.43) | (15.21) | (66.28) | (43.33) |
| LOO(3) | −7130.60 | −26375.10 | −33737.71 | −13345.55 | −8236.99 | −6747.46 | −37375.93 | −8716.61 |
| | (10.66) | (11.75) | (2.32) | (3.26) | (6.29) | (6.90) | (28.86) | (26.32) |
| LOO(3)+ | −7128.36 | −26369.93 | −33735.91 | −13346.03 | −8236.13 | −6751.97 | −37430.82 | −8691.38 |
| | (9.77) | (0.25) | (0.13) | (4.54) | (5.71) | (11.24) | (67.19) | (10.70) |
| $\mathcal{N}$(diag,4) | −7123.95 | −26382.91 | −33762.45 | −13341.62 | −8241.07 | −6735.78 | −37396.04 | −8679.48 |
| | (12.03) | (16.89) | (10.68) | (3.64) | (6.56) | (5.25) | (26.39) | (27.55) |
| LOO(3) | −7120.88 | −26368.53 | −33736.04 | −13338.99 | −8238.16 | −6735.59 | −37357.86 | −8665.54 |
| | (13.15) | (0.05) | (0.09) | (6.08) | (0.52) | (4.51) | (10.76) | (5.66) |
| LOO(3)+ | −7119.81 | −26368.49 | −33735.92 | −13339.79 | −8236.69 | −6736.74 | −37353.08 | −8665.99 |
| | (11.71) | (0.10) | (0.15) | (4.55) | (4.70) | (3.55) | (16.97) | (7.53) |
| $\mathcal{N}$(full,2) | −7129.70 | −26487.71 | −33807.05 | −13353.30 | −8251.01 | −6750.00 | −37487.49 | −8736.81 |
| | (6.14) | (54.79) | (22.97) | (5.92) | (10.34) | (11.91) | (50.43) | (52.38) |
| LOO(3) | −7132.35 | −26391.00 | −33736.98 | −13347.17 | −8237.75 | −6752.46 | −37462.07 | −8684.38 |
| | (6.89) | (11.88) | (1.89) | (7.77) | (6.08) | (8.64) | (54.40) | (7.98) |
| LOO(3)+ | −7122.76 | −26380.59 | −33736.93 | −13343.21 | −8239.96 | −6753.84 | −37419.02 | −8691.96 |
| | (10.81) | (14.39) | (2.10) | (2.14) | (4.84) | (14.30) | (35.94) | (13.51) |
| $\mathcal{N}$(full,4) | −7120.03 | −26378.55 | −33753.20 | −13342.27 | −8237.33 | −6734.51 | −37373.32 | −8662.53 |
| | (11.92) | (11.05) | (3.03) | (2.71) | (5.41) | (1.95) | (10.08) | (4.58) |
| LOO(3) | −7124.62 | −26368.49 | −33736.03 | −13337.74 | −8234.18 | −6734.49 | −37347.46 | −8666.63 |
| | (12.33) | (0.13) | (0.16) | (1.71) | (6.11) | (3.14) | (11.93) | (7.86) |
| LOO(3)+ | −7123.37 | −26368.51 | −33735.99 | **−13337.06** | −8241.25 | −6734.63 | −37352.30 | −8666.39 |
| | (11.28) | (0.09) | (0.05) | (1.45) | (8.15) | (2.18) | (12.32) | (7.54) |
| $\mathcal{WN}$(diag,2) | −7126.89 | −26444.84 | −33823.74 | −13358.16 | −8251.45 | −6745.60 | −37516.88 | −8719.44 |
| | (10.06) | (27.91) | (15.62) | (9.79) | (9.72) | (8.36) | (69.88) | (60.54) |
| LOO(3) | −7130.67 | −26380.41 | −33737.75 | −13346.94 | −8239.36 | −6741.63 | −37382.28 | −8690.41 |
| | (10.67) | (14.40) | (2.48) | (4.25) | (4.62) | (3.23) | (31.96) | (15.92) |
| LOO(3)+ | −7128.40 | −26375.28 | −33736.91 | −13347.32 | −8235.41 | −6742.40 | −37411.28 | −8683.22 |
| | (9.78) | (11.78) | (1.91) | (4.42) | (5.70) | (1.94) | (56.74) | (13.13) |
| $\mathcal{WN}$(diag,4) | −7122.10 | −26381.84 | −33759.19 | −13342.81 | −8243.92 | **−6733.38** | −37369.36 | −8666.85 |
| | (12.29) | (17.18) | (9.95) | (3.45) | (6.74) | (0.79) | (13.45) | (10.63) |
| LOO(3) | −7120.94 | −26368.52 | −33735.98 | −13339.77 | −8236.42 | −6735.12 | **−37341.92** | −8673.15 |
| | (13.11) | (0.03) | (0.08) | (3.84) | (3.63) | (2.52) | (9.15) | (0.97) |
| LOO(3)+ | −7125.78 | −26368.51 | −33736.00 | −13342.38 | −8235.03 | −6736.20 | −37345.80 | −8666.68 |
| | (13.10) | (0.10) | (0.18) | (6.35) | (5.36) | (1.91) | (11.13) | (5.78) |
| $\mathcal{WN}$(full,2) | −7124.63 | −26458.50 | −33804.63 | −13358.16 | −8251.09 | −6748.78 | −37484.98 | −8717.27 |
| | (8.09) | (32.42) | (20.76) | (1.20) | (7.46) | (7.38) | (34.91) | (28.49) |
| LOO(3) | −7125.94 | −26391.02 | −33736.98 | −13344.13 | −8236.90 | −6753.86 | −37416.00 | −8684.90 |
| | (13.07) | (11.89) | (1.96) | (0.22) | (5.13) | (10.68) | (3.12) | (12.81) |
| LOO(3)+ | −7115.19 | −26385.21 | −33736.97 | −13343.95 | −8239.55 | −6747.61 | −37431.76 | −8683.54 |
| | (8.16) | (13.77) | (1.93) | (1.76) | (4.72) | (6.87) | (43.65) | (3.57) |
| $\mathcal{WN}$(full,4) | **−7111.55** | −26379.48 | −33757.79 | −13342.71 | −8240.87 | −6735.14 | −37377.86 | −8663.51 |
| | (0.07) | (11.60) | (8.07) | (1.61) | (9.80) | (2.64) | (29.48) | (6.85) |
| LOO(3) | −7119.77 | **−26368.44** | −33736.01 | −13339.26 | −8234.06 | −6733.91 | −37350.77 | −8671.32 |
| | (11.80) | (0.13) | (0.03) | (3.19) | (7.53) | (0.57) | (11.74) | (5.99) |
| LOO(3)+ | −7116.09 | −26368.54 | **−33735.85** | −13337.42 | **−8233.89** | −6735.90 | −37358.96 | **−8660.48** |
| | (10.67) | (0.12) | (0.12) | (1.32) | (6.63) | (1.13) | (13.06) | (0.78) |

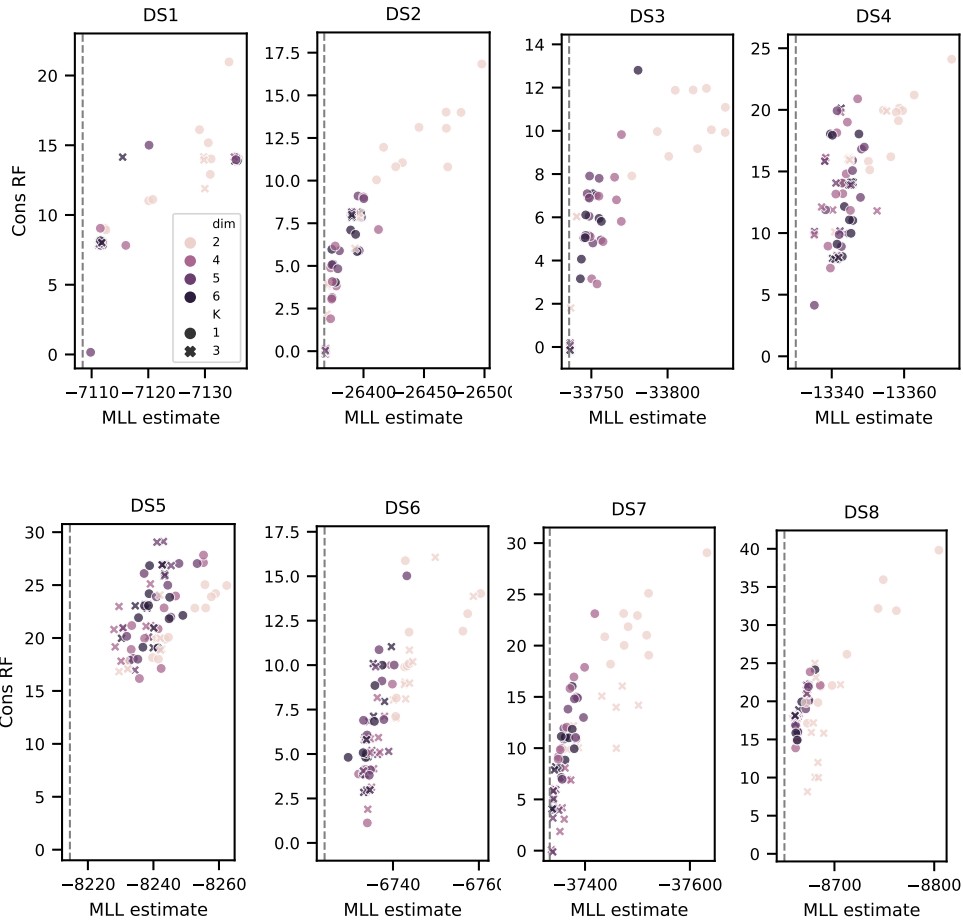

Figure 7: Marginal log-likelihood (MLL) estimates and Robinson-Foulds (RF) distance between the consensus trees obtained from topology distribution $Q(\tau)$ and MrBayes (MCMC) for datasets DS1-DS8. Each dot denotes an individual experimental run. The term 'dim' refers to the dimension of wrapped normal distributions used for $Q(z)$. To avoid dot overlap in the plot, RF metrics are randomly jittered within $\pm 0.2$. The dashed line illustrates the MLL value estimated by MrBayes.

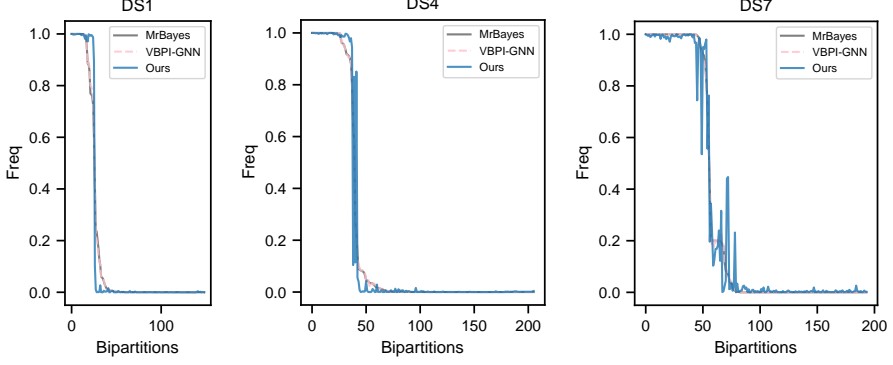

Figure 8: Comparison of bipartition frequencies from the posterior tree topology distributions of MrBayes, VBPI-GNN, and GeoPhy (ours), for DS1, DS4, and DS7 datasets. The bipartitions were ordered by descending frequency as seen in MrBayes.

Table 4: Comparative results of tree topology diversity statistics for DS1, DS4, and DS7 datasets using MrBayes, VBPI-GNN, and GeoPhy. Statistics presented include (a) the Simpson's diversity index, (b) the frequency of the most frequent topology, and (c) the number of topologies accounting for the top 95% cumulative frequency.

| Dataset | Statistics | MrBayes | VBPI-GNN | GeoPhy |
|---------|-----------|---------|----------|--------|
| DS1 | (a) | 0.874 | 0.891 | 0.362 |
|     | (b) | 0.268 | 0.313 | 0.793 |
|     | (c) | 42 | 44 | 11 |
| DS4 | (a) | 0.895 | 0.895 | 0.678 |
|     | (b) | 0.277 | 0.285 | 0.554 |
|     | (c) | 208 | 90 | 58 |
| DS7 | (a) | 0.988 | 0.991 | 0.996 |
|     | (b) | 0.022 | 0.028 | 0.022 |
|     | (c) | 753 | 315 | 553 |

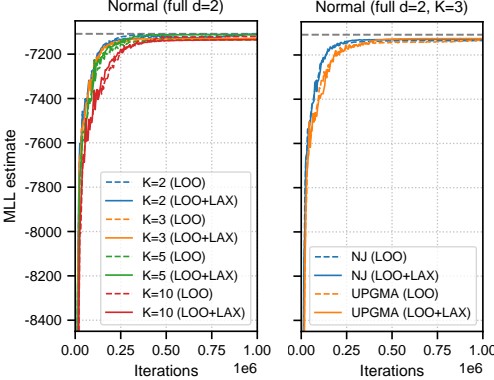

Figure 9: **Left**: Comparison of convergence curves for different per-step MC samples ($K = 2, 3, 5, 10$) in dataset DS1. **Right**: Comparison of convergence in dataset DS1 for different link functions, NJ and UPGMA, that map coordinates into tree topologies $\tau(z)$.

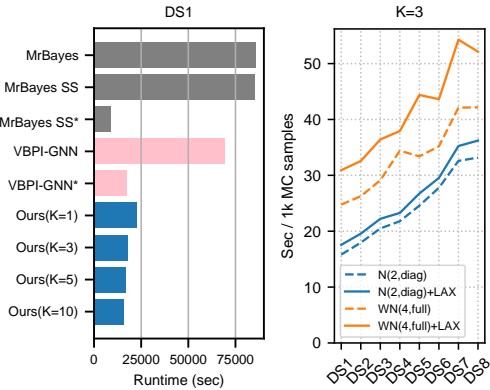

Figure 10: **Left**: Single-core runtime comparison of MrBayes, VBPI-GNN, and GeoPhy (ours). MrBayes SS* represents one of ten runs used for MrBayes SS. VBPI-GNN is trained for 400k iterations with $K = 10$ MC samples. For VBPI-GNN* and GeoPhy (ours), we represent runtimes for training with 1M MC samples. For GeoPhy, we employed a 2d-diagonal normal distribution for $Q(z)$ and LOO+LAX for control variates (CVs). **Right**: CPU-time per 1k MC samples for two types of distribution $Q(z)$ across eight datasets. We compared LOO or LOO+LAX for CVs.

