# OpenReview forum: "GeoPhy: Differentiable Phylogenetic Inference via Geometric Gradients of Tree Topologies"
_NeurIPS.cc/2023/Conference — NeurIPS 2023 poster_

### Official Review · Reviewer_Wg7V · 2023-06-25

**Soundness:** 3 good
**Presentation:** 4 excellent
**Contribution:** 4 excellent
**Rating:** 7
**Confidence:** 4

**Summary:**

This work presents a new robust and scalable method for inferring phylogenetic trees based on (variational) Bayesian inference.

**Strengths:**

Originality
- the originality of this work is in providing a robust and rigorous solution to an important application problem where the application of Bayesian methodology has so far been relatively limited

Quality
- the work is well motivated and appropriately implemented including the relevant derivations and experimental benchmarking

Clarity
- language is fluent, and intuitive illustrations support reading; the technical details are described in sufficient detail and complemented by intuitive explanation. Conclucions and limitations are clearly stated.

Significance
- significant, general and timely application problem addressed
- improved robustness, while maintaining scalability that allows practical application




**Weaknesses:**

Treatment of the benchmark data sets is limited compared to the potential of the method for real applications. This could be expanded to highlight the relevance of the work.

**Questions:**

Good performance compared to alternatives is shown but is it possible to demonstrate more on practical relevance for this?

**Limitations:**

No source code.

No assessment of societal implications.

---

> ### Author Rebuttal · Authors · 2023-08-10
>
> > 1. Treatment of the benchmark data sets is limited compared to the potential of the method for real applications. This could be expanded to highlight the relevance of the work.
>
> Thank you for your constructive comments.
> Regarding the improved treatment of the current experiments, we introduced an intuitive understanding and comparison of topology spread and consensus trees through visualization analysis, as shown in Fig. R1. Additionally, to gain a deeper understanding of the estimated topology distribution Q(τ), we analyzed the relationship between the marginal log-likelihood (MLL) estimates and the fidelity of the majority consensus tree obtained from $Q(\tau)$ in Fig. R2.
>
> > 2. Good performance compared to alternatives is shown but is it possible to demonstrate more on practical relevance for this?
>
> Thank you for your question. In Fig. R2, we confirm that the performance of the marginal log-likelihood (MLL) estimates aligns well with the quality of the consensus tree obtained from $Q(\tau).
> We believe that the estimation of evolutionary parameters and demographic history, which are more practical issues, will become possible in the future by expanding the simple phylogenetic tree model and deriving variational algorithms for it.
>
> > 3. No assessment of societal implications.
>
> Thank you for your insightful comments.
> Regarding the societal implication of our work,
> we recognize the importance of addressing this aspect.
> Scalable phylogenetic inference methods, like the ones we propose, have the potential to greatly enhance our understanding of the evolution, origin, and spread mechanisms of viruses and bacteria. Such insights could have profound societal impacts, especially in the areas of public health and disease control.
> We will include a discussion on the potential applications and the future perspectives related to these methods in the 'Limitations and Future work' section.
>
> > 4. No source code.
>
> We have included our source code in the supplementary materials.
> We will ensure that our updated manuscript provides clear references to it for more clarity.

---

> > ### Comment · Reviewer_Wg7V · 2023-08-15
> > **Review responses**
> >
> > Thank you, the comments have been adequately addressed. My overall scoring remains unaltered.

---

### Official Review · Reviewer_xhDC · 2023-06-29

**Soundness:** 4 excellent
**Presentation:** 3 good
**Contribution:** 2 fair
**Rating:** 3
**Confidence:** 2

**Summary:**

The authors propose a method for learning phylogenetic trees from sequence data.

The key idea is borrowed from [16], which is to represent the tree topology in terms of an embedding of the leafs of the trees in a continuous space, $z \in \mathcal{Z}$ from which the topology is extracted via a mapping $\tau \;:\; \mathcal{Z} \to \mathcal{T}$, where $\mathcal{T}$ is the space of binary trees, using a distance-based technique (in particular the popular Neighbor-Joining (NJ) method). The fact that the representation in terms of $z$ is continuous has benefits in a variational representation of the posterior distribution over trees.

The theoretical contribution of the present paper is resolving some issues with "an unevaluated Jacobian determinant between $B_\tau$ and $z$" (see Sec. 4 "Related work", p. 6).

The resulting method is demonstrated through simulations, showing that the method yields high-scoring models (high log-likelihoods), but not as high as existing variational and MCMC techniques (Table 2).

**Strengths:**

- I like the idea (which really borrowed from [16], so not really a new idea) of the embedding in terms of continuous coordinates
- solid theoretical derivation of the variational estimators
- promising results
- quite well written (although a bit dense)

**Weaknesses:**

- not clear what the benefits are compared to existing variational and MCMC methods (which are presented as the gold-standard)

**Questions:**

My main question is what are the benefits compared to, e.g., MrBayes, VBPI-GNN, and the method proposed in [33], the first two of which are referred to as gold standard. I'm guessing the proposed method is computationally more efficient and scalable, but this should be clearly demonstrated (or am I missing something?).

I should point out that I'm no expert of variational Bayes, so I didn't check the derivations from that point of view. I'm counting on the other reviewers to comment on that side.

detailed comments:
p. 7: "marginal log-likelihood (MLL) estimates": Does this mean MLL values? I mean, "maximum (log-) likelihood estimate" usually refers to an estimate of a parameter that is obtained by likelihood maximization, so it'd be good to be unambiguous.
p. 7: typo "Monte-Calro"


**Limitations:**

I think so (but see my question about the benefits wrt. existing methods above).

---

> ### Author Rebuttal · Authors · 2023-08-10
>
> Thank you for your constructive review and comments.
>
> > My main question is what are the benefits compared to, e.g., MrBayes, VBPI-GNN, and the method proposed in [33], the first two of which are referred to as gold standard. I'm guessing the proposed method is computationally more efficient and scalable, but this should be clearly demonstrated (or am I missing something?).
>
> Thank you for your question. Our main contribution is the development of variational Bayesian inference methods for phylogenetics without the preselection of tree topologies, which VBPI-GNN requires.
> For the evaluation of the computational efficiency, we have included the performance comparison in Fig. R3 First, where the runtimes of our methods are comparable to VBPI-GNN for the same number of likelihood evaluations.
> Although MrBayes is a highly optimized and practical implementation for MCMC-based Bayesian phylogenetic inference,
> the extensibility, variational Bayesian methods is a promising alternative that facilitates a gradient-based optimization, evaluation of model evidence, such as $\ln P(Y)$, based on a tractable approximate posterior distribution, and the extensibility of the algorithms to more complex and large models.
>
> > detailed comments: p. 7: "marginal log-likelihood (MLL) estimates": Does this mean MLL values? I mean, "maximum (log-) likelihood estimate" usually refers to an estimate of a parameter that is obtained by likelihood maximization, so it'd be good to be unambiguous.
>
> Thank you for your question. In the context of the variational inference, the marginal log-likelihood (MLL) estimate is used as a quality metric of the posterior approximation $Q(\tau, B_\tau)$.
> This is because the expectation value of MLL estimates $L^{(K)}[Q(\tau, B_\tau]$ is the lower bound of the true MLL values, which coincides when the $Q(\tau, B_\tau) = P(\tau, B_\tau | Y)$.
>
> > p. 7: typo "Monte-Calro"
>
> Thank you for your suggestions. We will correct the spelling of our revised manuscript.

---

### Official Review · Reviewer_Bru6 · 2023-07-05

**Soundness:** 3 good
**Presentation:** 4 excellent
**Contribution:** 4 excellent
**Rating:** 8
**Confidence:** 4

**Summary:**

The authors present a novel variational distribution for tree topologies, $Q(\tau)$, in the context of variational inference (VI) in phylogenetics. They construct their $Q(\tau)$ by introducing a continuous distribution $Q(z)$ in hyperbolic space and define $Q(\tau)$ by an expectation over the support of $Q(z)$ of an indicator function acting on a mapping from the hyperbolic space to the topology space; this allows sampling from $Q(z)$ and reconstructing \tau through the neighbour joining algorithm and, by the differentiable $Q(z)$ distribution, enables Monte-Carlo gradient estimation to maximize the Evidence Lower Bound (ELBO).

An ablation study is conducted to evaluate different $Q(z)$ distributions (Normal and Wrapped-Normal) and control variates. The model is then compared to other VI methods and MrBayes across 8 datasets.

**Strengths:**

The problem of phylogenetic tree reconstruction is paramount to understanding evolution, e.g., the evolution of species and cancer. The grand size of the tree topology space induced by a set of taxa constitutes a major obstacle for inference; any method successful in efficient exploration of this space is a significant contribution to the field. Here, the authors construct a differentiable $Q(\tau)$, enabling the use of gradients to indirectly maneuver the tree topology space. Furthermore, the proposed framework does not require restricting the tree topology space based on pre-processing steps as in VBPI.

The novel, differentiable approach to $Q(\tau)$, the avoidance of restrictions of the tree support and reporting strong results on datasets DS1-DS8 in terms of the ELBO(Q,R), makes the paper a significant and original contribution to the field of Bayesian phylogenetics.

**Weaknesses:**

The standard deviations of the proposed method reported in Table 2 are low for multiple datasets; this may be due to the consistency of the optimization across seeds, but may also indicate that the support of $Q(z)$ collapses to regions containing one/few unique $\tau(z)$. Since there is no analysis/discussion regarding this, the paper presents limited information regarding the representative power of $Q(\tau)$, which is a major set back of the paper, especially as the MrBayes "golden-run" posteriors over the tree topologies for DS1-DS8 support multiple topologies (Whidden and Matsen 2015, https://www.ncbi.nlm.nih.gov/pmc/articles/PMC4395846/ ).

The use of $Q(z)$ makes the method extra susceptible to limited topology support by the inherent mode-seeking behaviour in VI; while a mode in the topology space could represent multiple tree topologies, a mode in Z-space might only support reconstruction of, at worst, one tree topology.

The paper would benefit from reporting runtime compared to MrBayes and other VI methods, especially as one of the strengths in VI is speed. The current report on runtime is limited and is relegated to Appendix C.

Furthermore, an obvious weakness is the need to introduce a lower bound of the ELBO.

**Questions:**

The paper is a significant, novel and promising contribution to the field and is in its current format acceptable for publication.

However, based on my concerns in the weakness section, addressing the following points in the rebuttal could make me increase my score:
1. A discussion regarding the low variance in Table 2 and limited knowledge of the representative power $Q(\tau)$.
2. Clearer reporting of the runtime of the algorithm in the main paper.

Addressing the following points in the rebuttal could increase my score even further:
1. A experiment demonstrating the representative power/limitations of $Q(\tau)$


Some misprints: "Monte-Calro" figure text of Figure 2 line 4.

**Limitations:**

Limitations has been somewhat discussed (see weaknesses for omitted discussion of possible limitation).

---

> ### Author Rebuttal · Authors · 2023-08-10
>
> Thank you for your thoughtful suggestion and feedback.
>
> > W1. The standard deviations of the proposed method reported in Table 2 are low for multiple datasets; this may be due to the consistency of the optimization across seeds, but may also indicate that the support of $Q(z)$ collapses to regions containing one/few unique $\tau(z)$.
>
> > Q1-1. A discussion regarding the low variance in Table 2 and limited knowledge of the representative power $Q(\tau)$.
>
> Thank you for your question. We expect that the main source of relatively low variance in Table 2 of our methods comes from that the marginal log-likelihoods (MLLs) are sufficiently close to the reference values.
> Indeed, although VBPI-GNN needs the reasonable preselection of candidate topologies, it exhibits the lowest variance among the methods (Table 2) and still shows aligned bipartition frequencies to those of MCMC (Fig. R2 Fourth).
> The future work should address the expressivity of $Q(z)$ to represent a more diverse topological distribution $Q(\tau)$.
>
> > The paper would benefit from reporting runtime compared to MrBayes and other VI methods, especially as one of the strengths in VI is speed. The current report on runtime is limited and is relegated to Appendix C.
>
> > Q1-2. Clearer reporting of the runtime of the algorithm in the main paper.
>
> Thank you for your suggestion. We have compiled the runtime of our methods in comparison with MrBayes and VBPI-GNN (Fig. R3 First).
> While we expect that more efficient per-step computation will be an important future work, current performance for the fixed number of iterations is comparable to that of VBPI-GNN.
> We also included the estimated runtime across the eight datasets
> and the different $Q(z)$ model configurations,
> where the number of species ranges from $N=27$ to $64$ (Fig. R3 Second).
>
> > Q2-1. A experiment demonstrating the representative power/limitations of Q(τ)
>
> > The use of Q(z) makes the method extra susceptible to limited topology support by the inherent mode-seeking behaviour in VI; while a mode in the topology space could represent multiple tree topologies, a mode in Z-space might only support reconstruction of, at worst, one tree topology.
>
> Thank you for your constructive suggestion.
> We confirm that the mode of the topological distribution $Q(\tau)$ is well matched to that of MCMC when the MLL values are close to goldstandard, through the visualization of consensus trees obtained from $Q(z)$ and MCMC shown in Fig. R1 and
> the evident correlation between Robinson-Foulds (RF) distance and MLL estimates for multiple datasets in Fig. R2 First to Thrid.
> For $Q(\tau)$ in DS1 dataset, we showcase bipartition frequencies observed in the posterior tree distribution of MCMC, VBPI-GNN, and our methods (Fig. R2 Fourth).
> The divergence index of our $Q(\tau)$ in Fig. R2 Fourth is 0.36 > 0.
> However, the divergence is considerably lower than those of MCMC (0.87) and VBPI-GNN (0.86),
> and it is still difficult for our method to represent the tree distributions around the mode (the consensus tree).
>
> > Some misprints: "Monte-Calro" figure text of Figure 2 line 4.
>
> Thank you for your suggestion. We will correct the spelling.

---

> > ### Comment · Reviewer_Bru6 · 2023-08-17
> > **Response rebuttal**
> >
> > Thank you for addressing my concerns raised in the review. I will raise my score to a 7, for now, as the rebuttals added experiments and discussion adequately addresses Q1-1 and Q1-2. However, regarding Q2-1 I still have some questions. Furthermore, I was not aware of the resemblance to [16] pointed out by reviewer xhDC, which changes my assessment on novelty and therefore less prone to raise the score further.
> >
> > Q1-1:
> > I agree, this a plausible explanation for the low variance.
> >
> > Q1-2:
> > Figure R3 more than adequately addresses my concerns. In light of these results, the paper would benefit from mentioning the faster learning runtime of VCSMC and Vaiphy, illustrating the trade-off between high performance but long training runtime approaches (Geophy, VBPI) vs lower performance but short training runtime approaches (Vaiphy, VCSMC).
> >
> > Q2-1:
> > R2 First to third: I don't understand the experiment behind this, is the consensus trees of Geophy and MrBayes are calculated, and then the ELBO measured under this fixed tree setting? What do each dot represent in each plot? What do 'dim' refer to? Could you please provide a more elaborate description of the experiments behind this plot so I can better assess how they address my concerns.
> >
> > R2 Fourth:
> > I think that the DS1 dataset is not preferable for this analysis, as the support of the MrBayes posterior includes few topologies. Datasets DS4 and higher would've been a better selection (see figure 3 of https://www.ncbi.nlm.nih.gov/pmc/articles/PMC4395846/). Nonetheless, it does shine some light on the representative power of $Q(\tau)$ and is a good addition to the paper.
> >
> > Furthermore, what in R2 First-Fourth informs the diversity of the trees sampled by $Q(\tau)$? How many unique trees are used to construct the consensus-tree?

---

> > > ### Author Response · Authors · 2023-08-19
> > > **Response on the remaining concerns**
> > >
> > > We sincerely appreciate your detailed feedback and the time you've taken to review our manuscript.
> > >
> > > > I was not aware of the resemblance to [16] pointed out by reviewer xhDC, which changes my assessment on novelty and therefore less prone to raise the score further.
> > >
> > > We'd like to address the perceived resemblance between our work and [16] and clarify our contributions, which we partly discussed in the 'Related work' section.
> > > While we acknowledge [16] as an inspiring and original contribution to the field, we believe that our work also contributes distinctly and significantly.
> > >
> > > Similarities:
> > > * The use of the neighbor-joining (NJ) algorithm that maps continuous coordinates to a phylogenetic tree
> > > * Both works focus on Bayesian phylogenetic inference
> > >
> > > Differences:
> > > * We developed a variational inference (VI) algorithm, while [16] developed an MCMC-based algorithm.
> > > * We addressed the issue of parameterizing $Q(\tau)$ in the VI algorithm to cover a vast number of tree topologies, while [16] highlighted the fidelity of hyperbolic spaces to embed the distribution over trees with distances $P(\tau, B_\tau | Y)$.
> > > * For the use of the NJ, we explicitly defined a distribution over topologies $Q(\tau) = E_{Q(z)}[I[\tau(z) = \tau]]$ instead of mapping coordinates $z$ to $(\tau, B_\tau)$ as seen in [16]. This distinction is crucial as our approach avoids the issue of the Jacobian determinant seen in [16].
> > > * Unlike [16], our link function $\tau(z)$ does not necessarily rely on NJ, as we don't directly link $z$ to $B_\tau$. We showcase the results using UPGMA in Fig. R3 Fourth.
> > >
> > > Distinct contributions not comparable to [16]:
> > > * We introduced a tractable lower bound $\mathcal{L}$, then explored designs of variational distributions and control variates to complete a novel VI algorithm (GeoPhy).
> > > * We benchmarked the model evidence estimations (MLLs) across approaches and exhibited significant improvement over other methods that considered whole topologies.
> > >
> > > We hope these clarifications address your concerns.
> > >
> > > > the paper would benefit from mentioning the faster learning runtime of VCSMC and Vaiphy ...
> > >
> > > Thank you for your suggestion. Accordingly, we will include a discussion on the trade-off of the performance and runtimes between these methods in their standard use in our revised manuscript.
> > >
> > > > R2 Fourth: I think that the DS1 dataset is not preferable for this analysis, as the support of the MrBayes posterior includes few topologies.
> > > > Datasets DS4 and higher would've been a better selection
> > >
> > > Thank you for the valuable suggestions. In response, we've investigated DS4 and DS7 alongside DS1 to discuss the limitations of $Q(\tau)$, especially in cases with more diffused tree samples.
> > >
> > > Also, we present the difference of tree topology distributions more clearly by showing (b) the frequency of the most frequent topology and (c) the number of topologies up to their cumulative frequency matches to 95 percentile, in addition to (a) the diversity index.
> > >
> > > |  DS1 | MrBayes | VBPI-GNN | GeoPhy |
> > > |--|--:|--:|--:|
> > > | (a) Simpson's diversity index | 0.87 | 0.86  | 0.36 |
> > > | (b) Top freq. topology |  0.27 | 0.26 | 0.79 |
> > > | (c) #topology up to 95% freq. | 42 | 44 | 11 |
> > >
> > > | DS4 | MrBayes | GeoPhy |
> > > |--|--:|--:|
> > > | (a) | 0.90 | 0.68 |
> > > | (b) |  0.28 | 0.55 |
> > > | (c) | 208 | 58 |
> > >
> > > | DS7 | MrBayes | GeoPhy |
> > > |--|--:|--:|
> > > | (a) | 0.99 | 0.99 |
> > > | (b) |  0.02 | 0.02 |
> > > | (c) | 753 | 553 |
> > >
> > > For DS4, the overall tendencies in GeoPhy: lower (a), higher (b), and lower (c), are observed as DS1. Interestingly, for DS7, we observed that GeoPhy also represents more diverse tree samples than DS1 and DS4. However, the number of unique topologies up to 95% freq. is still lower than MrBayes, which implies the requirement of more expressiveness on $Q(\tau)$ to represent fine topology weights.
> > > While the results for VBPI-GNN are not readily available for DS4 and DS7 due to time constraints, we would like to include the corresponding results in our revised manuscript.
> > >
> > > > Furthermore, what in R2 First-Fourth informs the diversity of the trees sampled by $Q(\tau)$?
> > >
> > > Given that most tree topologies $\tau$ had very low frequencies, we used the bipartition frequencies of species defined for each tree topology edge as a more concise statistic for the distribution of $Q(\tau)$.
> > >
> > > In Fig. R2, we present bipartitions ordered by descending frequency as seen in MrBayes. The intermediate values between 0 and 1 in this frequency plot reveal topology diversities, where VBPI-GNN aligns more closely with MrBayes for the DS1 dataset compared to GeoPhy. As GeoPhy tends to take values near zero and one in the slope region, it indicates a need for increased expressiveness of $Q(\tau)$ to better represent intermediate frequencies. This trend was also noticeable for DS4. For DS7, while GeoPhy traced the curve more accurately, fluctuations around this curve highlight potential room for improvement. We intend to incorporate these figures in our updated manuscript.

---

> > > ### Author Response · Authors · 2023-08-19
> > > **Clarifications on Fig. R2 First to Third**
> > >
> > > We apologize for our oversight of your questions regarding Fig. R2 First to Third.
> > >
> > > > Q2-1: R2 First to third: I don't understand the experiment behind this, is the consensus trees of Geophy and MrBayes are calculated, and then the ELBO measured under this fixed tree setting? What do each dot represent in each plot? What do 'dim' refer to? Could you please provide a more elaborate description of the experiments behind this plot so I can better assess how they address my concerns.
> > >
> > > > How many unique trees are used to construct the consensus-tree?
> > >
> > > We complement the details for Fig. R2 First to Third as follows:
> > >
> > > Each dot represents a different experimental run. The term 'dim' stands for the dimension of wrapped normal distributions employed for $Q(z)$. We extended our experiments to include 5 and 6 dimensions in response to a query from  reviewer AJKq.
> > > The procedure is outlined below:
> > >
> > > 1. We sampled 1000 trees from the posterior distribution $Q(\tau)$ of GeoPhy
> > > 2. We then deduced the majority-rule consensus tree from these sampled trees
> > > 3. Subsequently, we calculated the Robinson-Foulds (RF) distances between the GeoPhy consensus tree and the consensus tree derived from MrBayes tree samples.
> > >
> > > Note that the RF metrics are randomly jittered within $\pm 0.2$ to prevent the overlapping of dots in the plot.
> > >
> > > As the consensus tree is derived from multiple sampled trees, the RF distance represents a summarized metric of alignment of $Q(\tau)$ to MCMC samples. Through Fig. R2 First to Third, we have confirmed that the performance metrics (i.e., MLL estimates) of the GeoPhy model are well aligned with the proximity of $Q(\tau)$ and the MCMC samples in terms of their consensus trees (the modes of the tree topology distribution).

---

> > > > ### Comment · Reviewer_Bru6 · 2023-08-21
> > > > **Regarding response to remaining concerns and clarification**
> > > >
> > > > Thank you for your extensive investigations regarding my remaining concerns. This, together with the clarifications, addresses all questions I had regarding the submission.
> > > > The added experiments demonstrate a limitation of the representative power of $Q(\tau)$ of Geophy w.r.t. e.g. VBPI. However, the investigation only makes the paper stronger as it suggests where further improvements of the method can be done. I will raise my score to an 8.

---

### Official Review · Reviewer_MJWZ · 2023-07-07

**Soundness:** 3 good
**Presentation:** 3 good
**Contribution:** 4 excellent
**Rating:** 7
**Confidence:** 4

**Summary:**

Authors proposed GeoPhy as a fully differentiable approach for phylogenetic inference, addressing a fundamental problem in phylogenetic inference. In experiments with real benchmark datasets, GeoPhy demonstrated its superior performance compared to other methods when considering all topological candidates. This approach is of interest to the general ML community.

**Strengths:**

Overall, the manuscript is well-written.

**Weaknesses:**

It is highly recommended to include an algorithm block summarizing the GeoPhy method from input to output for clarity. Additionally, there are some details in the experimental section that need clarification.

**Questions:**

Major questions:

1. In Section 3.1, the neighbor-joining algorithm is defined as the projection from the distance matrix to the tree topology. However, other standard algorithms, such as UPGMA, could be alternative approaches. Have you investigated if the choice of projection leads to different performance in the benchmark tests?

2. Appendix C.5 provides a rough estimation of the running time, but it is unclear how the other baseline models included in the benchmark tests perform in terms of time. Furthermore, how does the algorithm scale as the number of tips and sites increase? Have you observed any trends in the time curve with respect to the number of taxa and sites?

3. In Table 2, GeoPhy achieves superior performance in the benchmark tests based on likelihoods. Did the optimized topologies look significantly different across the different methods?

4. How were the per-step Monte Carlo samples (K) determined in the benchmark tests? There is no clear winner based on Table 1.

5. Could you elaborate on the approaches used to calculate the marginal log-likelihood (MLL) for all the listed methods?

**Limitations:**

Authors briefly mentioned two limitations of the proposed model: 1) Q(z) can be improved to enhance its expressive power, and 2) the computation cost per update step can be optimized to speed up the overall inference process. These practical concerns can be further addressed in future studies.

---

> ### Author Rebuttal · Authors · 2023-08-10
>
> > W1. It is highly recommended to include an algorithm block summarizing the GeoPhy method from input to output for clarity. Additionally, there are some details in the experimental section that need clarification.
>
> Thank you for your suggestion. We will include the algorithm block that summarizes the method workflow in our revised manuscript.
>
> > Q1. In Section 3.1, the neighbor-joining algorithm is defined as the projection from the distance matrix to the tree topology. However, other standard algorithms, such as UPGMA, could be alternative approaches. Have you investigated if the choice of projection leads to different performance in the benchmark tests?
>
> Thank you for your suggestion. Our framework defined in equation (4) certainly allows an alternative link function to obtain a tree topology.
> We have included an experiment with UPGMA with a promising optimization trajectory (Fig. R3 Fourth).
>
> > Q2. Appendix C.5 provides a rough estimation of the running time, but it is unclear how the other baseline models included in the benchmark tests perform in terms of time. Furthermore, how does the algorithm scale as the number of tips and sites increases? Have you observed any trends in the time curve with respect to the number of taxa and sites?
>
> Thank you for your suggestion.
> we have analyzed the runtime of MrBayes for MCMC and Stepping-Stone (SS) method, VPBI-GNN, and our methods with their standard use we have employed in this work for dataset DS1 (Fig. R3 First).
> As MrBayes is written in C and highly optimized its per-iteration computation time is much faster than current variational methods.
> Runtimes of our methods per iteration (number of likelihood evaluations) were comparable to VBPI-GNN.
>
> > Q3. In Table 2, GeoPhy achieves superior performance in the benchmark tests based on likelihoods. Did the optimized topologies look significantly different across the different methods?
>
> In our additional experiments, we have confirmed that the discrepancy measure (Robinson-Foulds distance) of the majority-rule consensus trees derived from the topological distribution $Q(\tau)$ and MCMC (MrBayes) are highly correlated with the marginal log-likelihood (MLL) estimates (Fig. R2 First to Third).
> We have not evaluated topology distributions obtained with methods whose MLL estimates were far deviated from the reference values.
> Although when the consensus tree of our inference runs exactly matches that of MCMC, it is still difficult to express the distribution of tree topologies faithfully due to the limitation of expressivity in $Q(z)$.
> We exemplify the case with DS1 dataset in Fig. R2 Fourth,
> where VBPI-GNN shows more aligned bipartition frequencies to MrBayes than ours.
>
> > Q4. How were the per-step Monte Carlo samples (K) determined in the benchmark tests? There is no clear winner based on Table 1.
>
> Thank you for your question.
> We have observed that the smaller $K$ shows relatively faster convergence in terms of the number of iterations.
> We included the corresponding experiment in Fig. R3 Third.
>
> > Q5. Could you elaborate on the approaches used to calculate the marginal log-likelihood (MLL) for all the listed methods?
>
> For VBPI-GNN, the importance-weighted ELBO with 1000 iterations was used for the MLL estimates.
> For CSMC-based methods, we will include the details of computations in our revised manuscript.

---

> > ### Comment · Reviewer_MJWZ · 2023-08-21
> >
> > I have read through the responses and all of my concerns have been properly addressed. Thus I raised my score to 7.

---

### Official Review · Reviewer_AJKq · 2023-07-18

**Soundness:** 3 good
**Presentation:** 3 good
**Contribution:** 3 good
**Rating:** 5
**Confidence:** 5

**Summary:**

This paper proposed a family of implicit distribution over tree topologies which allows support free variational Bayesian phylogenetic inference. The distribution is constructed based on the neighbor joining algorithm which maps a distribution over the tip node vectors to the tree topology space. Both Euclidean and hyperbolic spaces are considered for the tip node distributions. As the resulting distribution over tree topologies is intractable, an auxiliary reverse distribution was introduced which leads to a looser lower bound for optimization. Various gradient estimators were investigated for the tree topology variational parameters optimization. Experiments on a benchmark of real data problems demonstrated the advantage over other variational approaches (mainly sequential monte carlo based) that does not require preselected tree topololgies.

**Strengths:**

1. The paper is written clearly and well organized. The problem of constructing flexible families of distributions over tree topologies is important for the phylogenetic communities and would to of interest to many practitioners.
2. In addition to a careful derivation to the training objective, the authors also investigated different gradient estimators for tree topology variational parameters.

**Weaknesses:**

1. As admitted by the authors, the proposed variational approximation $Q(z)$ and the reverse distribution $R(z|\tau)$ is simple, which may damage the overall approximation quality of the method.
2. In the derivation of the lower bound, the reverse distribution $R(z|\tau)$ needs to satisfy the constraint: the support of $R(z|\tau)$ should be inside the region $\{z: \tau(z)=\tau\}$.
2. In the experiments, it seems that the performance would be really sensitive to the choice of gradient estimators. I wonder if that is related to the small sample size $K$? The marginal likelihood estimates also tends to have large variance in most cases.
3. It may be hard to assess the performance on the tree topology posterior estimation as the distribution is implicit and defined in a rather sutble way.

**Questions:**

1. In equation 10 and equaiton 12, the expection of the last term is actually zero. Any idea why this term is kept there given that other control variates are considered?
2. How many samples are used to compute the marginal likelihood estimates?
3. The current construction of the reverse distribution $R(z|\tau)$ seems not take the constraints $\tau(z)=\tau$ into consideration directly. Would this have a negative impact on performance?
4. I note that only low dimension tip node vectors ($d\leq 4$) are considered in this paper. Would higher dimension embedding be helpful?
5. How about the tree topology approximation given by the implicit distribution? Although it may be hard to estimate the density, one can also report summary statistics from the sampled trees, for example.

**Limitations:**

yes, they do.

---

> ### Author Rebuttal · Authors · 2023-08-10
>
> Thank you for your thoughtful review and feedback.
>
> > W1. As admitted by the authors, the proposed variational approximation $Q(z)$ and the reverse distribution $R(z | τ)$ is simple, which may damage the overall approximation quality of the method.
>
> We believe that enhancing the expressiveness of $Q(z)$ and $R(z | \tau)$ is a valuable and non-trivial direction for future work. However, despite the simplicity of our choice, we consider it a significant contribution that we can consistently outperform CSMC-based methods (Table 3).
>
> > W3. In the experiments, it seems that the performance would be really sensitive to the choice of gradient estimators. I wonder if that is related to the small sample size K? The marginal likelihood estimates also tends to have large variance in most cases.
>
> Thank you for your detailed feedback.
> While the variance of our methods is still higher than that of goldstandard MCMC runs,
> we have confirmed that we can achieve better optimization than CSMC-based methods in most cases
> by using either $K=1$ (LAX), $K=3$ (LOO), or $K=3$ (LOO+LAX) (Table 2). We are interested in whether increasing
> $K$ could help reduce variance. However, we observed that increasing $K$ requires more iterations to achieve the same accuracy (Fig. R3 Third).
>
> > Q1. In equation 10 and equaiton 12, the expection of the last term is actually zero. Any idea why this term is kept there given that other control variates are considered?
>
> Thank you for your question.
> It may be confusing because we refer to $\nabla_\theta Q_\theta(z)$ as the score function.
> While the expectation of $\nabla_z \log Q_\theta(z)$ is zero, the term with $\nabla_theta$ remains.
> We will include a note in the footnote for clarification.
>
> > Q2. How many samples are used to compute the marginal likelihood estimates?
>
> We used 1000 MC samples. We will move the description from Table 1 caption to section 5.1 Experimental setup for more clarity.
>
> > W2. In the derivation of the lower bound, the reverse distribution $R(z | \tau)$ needs to satisfy the constraint: the support of $R(z | \tau)$ should be inside the region $z: \tau(z)=\tau$.
> > Q3. The current construction of the reverse distribution
>  seems not take the constraints
>  into consideration directly. Would this have a negative impact on performance?
>
> Thank you for highlighting this important point.
> Initially, we believed that the expressiveness of $R$ is only beneficial when a more complex $Q$ is used.
> At least given that $L[Q] \geq L[Q, R]$,
> it is not expected that we overestimate the quality of $Q$.
> However, there might be cases like
> $L[Q] \geq L[Q'] \geq L[Q', R] \geq L[Q, R]$,
> so even with a simple $Q$,
> designing $R$ to meet the conditions could prevent suboptimal trnasitions like $Q \to Q'$.
> We would like to include the discussion in our revised manuscript.
>
> > Q4. I note that only low dimension tip node vectors (
> ) are considered in this paper. Would higher dimension embedding be helpful?
>
> Thank you for your question. There is an advantage in using a low dimension as it reduces the number of parameters. To answer the question empirically, we have added experiments for $Q(z)$ with dimensions 5 and 6 (Fig. R2 Left to Right). While the difference between dims 2 and 4 is more pronounced, there appears to be a slight improvement in DS3 and DS7 when the dimension is increased to 5 or 6.
>
> > W4. It may be hard to assess the performance on the tree topology posterior estimation as the distribution is implicit and defined in a rather sutble way.
> > Q5. How about the tree topology approximation given by the implicit distribution? Although it may be hard to estimate the density, one can also report summary statistics from the sampled trees, for example.
>
> As we can sample easily from $Q(z)$, sampling
> from $Q(\tau)$ is straigtfoward.
> Therefore, evaluating the empirical distribution is not an issue.
> Each of the consensus tree used in Fig. R1 and R2 was obtained from 1000 samples obtained from $Q(\tau)$

---

> > ### Comment · Reviewer_AJKq · 2023-08-16
> > **Thanks for the reply**
> >
> > Thanks for the reply. The issue with equation 10 and 12 still remains. I think the current form in equation 10 is wrong, the remaining term from the gradient of the entropy is not the last term in equation 10 (which is the score and hence has 0 expectation). As the author admitted, the reverse distribution $R(z|\tau)$ does not take the constraint into consideration, which would lead to potential problems, e.g., the lower bound is not right. The samples from $Q(\tau)$ do not seem to provide good approximation to the exact posterior, this would be a cause of the biased estimate of MLL compared to MrBayes/VBPI-GNN. I will keep the score.

---

> > > ### Author Response · Authors · 2023-08-17
> > > **Clarifications on review points**
> > >
> > > Thank you for your continued feedback and assessments.
> > >
> > > > The issue with equation 10 and 12 still remains. I think the current form in equation 10 is wrong, the remaining term from the gradient of the entropy is not the last term in equation 10 (which is the score and hence has 0 expectation)
> > >
> > > We acknowledge an oversight in our previous rebuttal wherein we mistakenly wrote $\nabla_\theta Q_\theta(z)$ instead of $\nabla_\theta \ln Q_\theta(z)$.
> > > It's worth emphasizing that both $\nabla_z \ln Q_\theta(z)$ and $\nabla_\theta \ln Q_\theta(z)$ are referred to as the 'score function' in literature, but we meant the latter in our manuscript. Notably, while $E_{Q_\theta(z)}[ \nabla_z \ln Q_\theta(z) ] = 0$, the expectation $E_{Q_\theta(z)}[\nabla_\theta \ln Q_\theta(z) ]$ is not zero in general.
> > >
> > > Regarding the transformation of $\mathbb{H}[Q_\theta(z)]$ in Equation 10, we utilized the reparameterization trick as follows:
> > >
> > > $
> > > \nabla_\theta \mathbb{H}[Q_\theta(z)]
> > > = - \nabla_\theta E_{Q_\theta (z)} [ \ln Q_{\theta}(z) ]
> > > = - \nabla_\theta E_{p_z(\epsilon_z)}[ \ln Q_{\theta}(h_\theta(\epsilon_z)) ]
> > > = - E_{p_z(\epsilon_z)} [ \nabla_{\theta} \ln Q_{\theta}(h_\theta(\epsilon_z)) ].
> > > $
> > >
> > > This aligns with the second term of Equation 12.
> > > We will include the transformation above in our revised manuscript for more clarity.
> > > We hope this addresses the concerns raised.
> > >
> > >
> > > > As the author admitted, the reverse distribution $R(z | \tau)$ does not take the constraint into consideration, which would lead to potential problems, e.g., the lower bound is not right. The samples from $Q(\tau)$ do not seem to provide good approximation to the exact posterior, this would be a cause of the biased estimate of MLL compared to MrBayes/VBPI-GNN.
> > >
> > > We would like to complement the above arguments as follows:
> > >
> > > * The optimization target $L[Q, R]$ is still a right lower bound that satisfies $\ln P(Y) \geq L[Q] \geq L[Q, R]$, irrespective of constraints on $R(z | \tau)$.
> > > * However, the maximization of $L[Q, R]$ with respect to $Q$ might yield a suboptimal $Q$ if $L[Q, R]$ hasn't been fully maximized with respect to $R$.
> > > * Importantly, maximizing $L[Q, R]$ with respect to $R$ will elicit $R$ to satisfy the constraint (i.e., all $z$ in the support of $R(z | \tau)$ satisfies $\tau(z) = \tau$). Thus, it is not imperative to modify our framework to explicitly ensure the constraint for $R$ (except the expressiveness of $R$ to be able to meet the condition).
> > > * As we partly discussed in the section "Limitations and Future work", the remaining challenge for further improvement on the quality of posterior approximation $Q(\tau)$ is the introduction of more expressive distribution families for $Q$ as well as $R$.
> > >
> > > We appreciate the reviewer to shed light on these important points. We will address these points in detail in our revised manuscript.

---

> > > > ### Comment · Reviewer_AJKq · 2023-08-17
> > > >
> > > > Thanks for the clarification on equation 10 and the validity of the lower bound irrespective of the constraints on $R(z|\tau)$. I would suggest a construction of reverse model that satisfies the constraints if possible.

---

> > > > > ### Author Response · Authors · 2023-08-17
> > > > > **Response on constraints for the reverse model**
> > > > >
> > > > > Thank you for the confirmation of the above clarifications.
> > > > >
> > > > > > I would suggest a construction of reverse model that satisfies the constraints if possible.
> > > > >
> > > > > Thank you for your suggestion. We agree that directly introducing constraints on the reverse model $R(z | \tau)$ would be valuable, as it could simplify and improve the optimization process. Unfortunately, we suppose that it is not easy to define a family of probability densities for $R(z | \tau)$ so that the constraint is always satisfied because, to our knowledge, neither the boundary nor the area of such support is known for arbitrary $\tau$. However, as we present in additional experiments (Fig. R3 Fourth), it is possible to perform our approach with a link function $\tau(z)$ other than that based on the neighbor-joining (NJ) algorithm. Therefore, it might be also possible to design another link function $\tau(z)$ so that the domain for $\tau(z) = \tau$ is easier to specify. Exploring this avenue could be an interesting direction for future work.

---

### Author Rebuttal · Authors · 2023-08-10

We thank all the reviewers for taking the time to provide thorough and insightful feedback on our manuscript.
Your constructive comments have greatly enhanced the quality of our work.
In response to the points raised, we have engaged in a detailed discussion on the expressivity, performance, and limitations of our study with Figures R1 to R3 in the attached PDF file.

---

### Decision · Program_Chairs · 2023-09-21

**Decision:**

Accept (poster)

**Comment:**

Thank you for engaging with the reviewers in the discussion. The reviewers and I are mostly in agreement that this paper should be accepted, given the the interesting methodology and clear exposition. However, the reviewers expressed some concerns, and after reading and considering the discussion I echo a number of these:

- Reviewer AJKq noticed that the derivation of the lower bound is only correct when $R(z|\tau)$ generates draws that respect the constraints of the tree construction. More technically, we require that $R(z|\tau)$ *dominates* $Q(z|\tau)$ for $KL(Q||R)$ to be well defined. The problem is that $R(z|\tau)$ is a distribution with full support on $z$, while $Q(z|\tau)$ only has support on a degenerate space where $\tau = \tau(z)$. Based on the authors' responses, I was not sure whether this issue was fully appreciated. However, at the end of the math display after line 141, the final expression just involves the ratio of $Q(z)$ and $R(z | \tau)$, which should be fine (both likely have full support, and are continuous densities without atoms). The authors should ideally find a proof of proposition 1 that doesn't involve a dubious step with the (probably) ill-defined KL. In the worst-case, the authors should clearly point out this limitation.
- The paper should be clear about its connection to [16] and standard variational lower bounds / stochastic gradient optimization.
- The effect of the control variates in Table 1 and 2 is somewhat inconclusive. The variation of results with respect to the various control variates is small compared to its performance with respect to CSMC, VCSMC, and $\phi$-CSMC. It would probably clarify the paper to just pick one any of these and defer this comparison to the appendix.
- The experiments section should provide an unbiased and accurate assessment of the results. In particular, the comparison to VBPI (and MrBayes) is underwhelming. Both VBPI and MrBayes generally outperform the method in terms of marginal log-likelihood bound. The authors argue that VBPI requires tree pre-selection, which is true, but the argument is not very strong because VBPI uses automated methods for the pre-selection step.
- The experiments in the submitted draft only show MLL comparisons, but not anything more particular (recovery of MCMC trees, for example). The authors did address this issue in the rebuttal with RF comparisons, but note that these results seem to show that VBPI/MrBayes are again competitive. The camera ready paper should include an unbiased assessment of these results.

Please make sure to explicitly address these issues in the camera ready.